# Mass Timber Envelopes in Passivhaus Buildings: Designing for Moisture Safety in Hot and Humid Australian Climates

**Marcus Strang [1,*], Paola Leardini [1], Arianna Brambilla [2] and Eugenia Gasparri [2]**

1    School of Architecture, The University of Queensland, Brisbane 4072, Australia; p.leardini@uq.edu.au
2    School of Architecture, Design and Planning (CoCoA Lab), The University of Sydney, Sydney 2006, Australia; arianna.brambilla@sydney.edu.au (A.B.); eugenia.gasparri@sydney.edu.au (E.G.)
\*    Correspondence: m.strang@uq.net.au

**Abstract:** The uptake of buildings employing cross-laminated timber (CLT) assemblies and designed to Passivhaus standard has accelerated internationally over the past two decades due to several factors including responses to the climate crisis by decarbonising the building stock. Structural CLT technology and the Passivhaus certification both show measurable benefits in reducing energy consumption, while contributing to durability and indoor comfort. However, there is a general lack of evidence to support a fast uptake of these technologies in Australia. This paper responds to the compelling need of providing quantitative data and adoption strategies; it explores their combined application as a potential pathway for climate-appropriate design of energy-efficient and durable mass timber envelope solutions for subtropical and tropical Australian climates. Hygrothermal risk assessments of interstitial condensation and mould growth of CLT wall assemblies inform best-practice design of mass timber buildings in hot and humid climates. This research found that the durability of mass timber buildings located in hot and humid climates may benefit from implementing the Passivhaus standard to manage interior conditions. The findings also suggested that climate-specific design of the wall assembly is critical for mass timber buildings, in conjunction with excellent stormwater management practices during construction and corrosion protection for metallic fasteners.

**Keywords:** cross-laminated timber; hygrothermal; energy; moisture; durability; tropical; passivhaus

## 1. Introduction

### 1.1. Mass Timber and the Passivhaus Standard for Decarbonising the Construction Sector

The past two decades have witnessed a global acceleration internationally in uptake of both mass timber buildings and the Passivhaus (PH) standard [1,2]. This acceleration is mostly due to the construction sector's response to the current climate crisis by decarbonising the building stock. Indeed, established environmental benefits of mass timber technologies, such as cross-laminated timber (CLT), include good thermal properties and reduced embodied $CO_2$ emissions, among others [3], while the PH standard primarily leads to energy efficiency improvements during the operational phase, this has additional benefits such as occupant comfort [4] and increases in productivity [5]. The paired use of mass timber for reducing embodied energy and the PH standard for reducing operational energy can assist in decarbonising the construction sector. This strategy can be particularly significant in Australia, which urgently needs to progress its sustainability agenda as its effort to improve energy-efficiency is ranked as the worst of any major developed country in the world [6].

### 1.2. The Australian Context

PH standard is a voluntary quality assurance standard focused on reducing operational energy demands of buildings to a very low level, while maximising their indoor air

quality and thermal comfort (as per ISO 7730 [7]) for occupants. To achieve compliance with the PH standard, measurable performance criteria must be met. This is typically achieved by highly energy-efficient envelope design, where the space heating and cooling demands are significantly reduced.

The standard has validity for all climates of the world [8] and the extremely high energy savings resulting from its application in heating prevailing climates have been proven in a reproducible manner through statistically significant empirical studies in a number of European cities [9]. The first PH building was constructed in Darmstadt, Germany in 1991, and it has been estimated that over 60,000 Passivhaus buildings have been constructed globally over the preceding 25 years [10]. Despite its growing international recognition, the PH standard's uptake rate by the Australian construction industry remains slow, mostly due to technical gaps, still emerging component markets, and additional upfront costs. The first certified PH in Australia was constructed in 2013 [11], and the number of PH-certified buildings in the country has grown to 35 over 8 years at the time of publication [12]. While there are no buildings certified to the Passivhaus standard in Australian hot and humid climates to date, a two-storey single family home in Brisbane [13], has recently received certification under the Passive House Institute (PHI) Low Energy Building Standard [14].

Slow uptake has similarly occurred for mass timber buildings in Australia, especially in hot and humid climates. This is clearly demonstrated by the time gap between the first Australian mass timber building with a CLT wall assembly, the Forté Apartments in Melbourne, which was completed in 2012 [15], and the NIOA facility in Brisbane, the first multi-storey building with a CLT wall assembly in a hot and humid Australian climate, which was only completed in 2021 [16]. This slow uptake is in part due to psychological barriers as building designers and developers in Australia still hold prejudices about the durability of mass timber in construction [17]. Therefore, knowledge and understanding of both, PH standard and CLT construction in hot and humid Australian climates remains limited.

As the Australian building industry slowly transitions towards high-performance buildings for net-zero carbon emissions, a greater level of insulation and lower levels of air infiltration are prescribed. For instance, a recent addition to the Australian National Construction Code (NCC) 2019 now allows a maximum fabric infiltration rate of 10 air changes per hour (ACH) at 50 Pa for buildings containing sole-occupancy units if complying through the JV4 building envelope sealing performance requirement [18]. As first experienced in Europe with the application of external thermal insulation composite systems [19], and later in Australia [20], early energy-efficient buildings employing sealing strategies with insufficient ventilation, and moisture-safety redundancy in the assembly may have the unintended consequence of increasing internal relative humidity and the incidence of condensation, when not correctly designed [21,22]. This condition may exacerbate an endemic problem of Australian buildings: a longitudinal study of Australian residential properties confirms that a lack of ventilation is a major cause of mould issues [23]. This impacts indoor air quality (IAQ), resulting in higher latent loads and increased humidity-factors that are more supportive of mould growth [24]. It has been estimated that a third of new and existing Australian buildings suffer from condensation problems and moisture defects, leading to remediation works and significant adverse health effects [25]. However, previous studies have not yet considered in detail hot and humid climates, nor highly energy-efficient performance buildings such as buildings certified to the PH standard.

Addressing the lack of research for highly energy-efficient building envelopes in hot and humid climate is therefore essential, and may inform correct CLT assembly details for durable, energy-efficient, and comfortable mass timber buildings in hot and humid Australian climates. The relevance of this topic is highlighted by Gasparri et al. [26], that found timber-based envelopes in Australian tropical and sub-tropical climates are highly critical, especially during the cooling season. These findings align with the international literature, which extensively documents the risks of vapour barriers on the internal mass

timber surface in hot and humid climates [26–29], confirming that an internal moisture-open approach allows for a faster drying process.

Results of studies on mass timber assemblies in highly energy-efficient buildings can also be adapted to similar subtropical and tropical climates around the world. This is especially relevant, as hot and humid climates are projected to geographically expand due to climate change [30], with one third of the growing global population expected to experience temperatures ranges outside of human's adaptive capacity [31]. Decreasing ambient comfort conditions, accompanied by decreasing air conditioning costs and population growth, are likely to dramatically increase global cooling demands [32]. Increased climatic loads, including solar radiation (UV and temperature), wind and precipitation also heighten risks for buildings constructed in hot and humid climates [33].

## 2. Aim and Scope

The aim of this paper is to expand the current knowledge of mass timber assemblies in subtropical and tropical climates, thereby informing climate-specific design. Gillies Hall student accommodation, which is Australia's largest Passivhaus building constructed from CLT and designed for Melbourne's temperate climate, is used in this study as a pilot project to verify the durability of an energy efficient mass timber building located in selected Australian hot and humid climates. Hygrothermal assessments evaluate the CLT wall assembly solutions to minimise mould risks, specifically in respect to the position of the insulation layer, weather resistive barrier (WRB), and influence of good stormwater management. A parametric study is performed to assess the capacity of the selected assemblies to control the growth of mould and rate of corrosion. Filling these knowledge gaps is essential to inform construction details for mass timber buildings that prove effective for durability, energy-efficiency, and comfort in hot and humid Australian climates.

## 3. Research Context

Building's envelopes are constructed from multi-layered assemblies, where each layer allows some level of control over the ambient outdoor conditions. Control layers should manage the movement of rainwater, air, vapour, and heat, in that order of importance [34]. The rain, air, and vapour control layers should be defined and implemented correctly for building durability, while the thermal control layer should be designed for energy savings and increases in thermal comfort. These four control layers must encapsulate the entirety of the building envelope. At each junction, the designer must understand how the control layers are kept continuous across the detail to avoid thermal bridging, air-leakages, and water ingress.

### 3.1. Highly Energy-Efficient Envelopes

The 'perfect wall' for energy-efficient structures has been described as having the rain, air, vapour, and the thermal control layers between the exterior insulation and the structural layer [34]. These should be designed to withstand temperature fluctuations, building movement, and wind pressure, while minimising exposure to UV, moisture, and contaminants. This benefits the assembly by way of protecting all the UV-sensitive materials with the rain-screen cladding, while all temperature-sensitive materials are protected by the thermal layer, and all the moisture-sensitive materials are protected by a water control layer. This is of benefit in a cold and temperate climate as the structure is kept close to interior temperatures due to the position of the thermal layer. While any moisture accumulation that may occur will dry to the exterior or interior under almost consistent drying conditions. In addition, as the air and vapour control layers are also kept close to interior temperature and are situated outside of the structure, the tendency for condensation to occur within the structure is reduced. This is because the dew point then occurs within the insulation layer which is often resistance or more tolerant of moisture [34].

However, the 'perfect wall' has not been confirmed for CLT construction in hot and humid climates. The design of any assembly must be climate specific; in cold climates, the

goal is to make it as difficult as possible for the building assemblies to become moist from the interior. Therefore, best practice construction in cold climates is to install the air control layer on the interior side of the thermal control layer, and a vapour permeable material on the exterior side of the insulation, allowing the building assemblies to dry to the exterior.

Hot and humid climates with long cooling periods present the opposite challenge; tropical regions often experience intense rain periods, thus methods should be considered to avoid exacerbation of corrosion, rust, and decay due to possible water infiltration in buildings. Airtightness is critical to prevent moist exterior air from leaking inwards and condensing within the envelope. In tropical climates, the external dew point temperature can be as high as 28 °C [35]; as conditioning systems in these climates often operate below the dew point, the vapour pressure difference across the building envelope of an air conditioned building is very high, encouraging moisture condensation on cooler surfaces [36]. To maintain the desired indoor conditions while avoiding interstitial condensation, the building fabric must resist vapour migration and heat flow towards the interior. To achieve this, the AIRAH DA20 technical manual for air conditioning, cooling and comfort in hot and humid tropical climates [35] suggests that vapour control barriers should be located on the outside facing surface of the thermal control layer. Additionally, building assemblies must be allowed to dry towards the interior by using vapour permeable interior wall finishes [37].

Along with the envelope, the building systems in hot and humid climates must also be climate specific, where the cooling system is designed for the high humidity and latent loads. Controlled ventilation is an integral part of the building systems for suppling fresh air in an airtight envelope, such as required for Passivhaus-certified buildings. This allows mechanical systems in Passivhaus-certified buildings to avoid over-sizing the capacity of the systems. Typically, controlled fresh air ventilation is ensured using heat recovery units, which also reduces space cooling energy consumption. Enthalpy wheels in the mechanical ventilation system may be considered to recover latent heat energy, while a dehumidification system may further ensure indoor critical relative humidity levels are not exceeded.

Similarly, to the climate-specific requirements of the mechanical systems, the thermal performance of the envelope is also automatically assessed for climate appropriateness through the Passivhaus design process. Utilising mass timber in the building envelope not only leads to environmental benefits [38], but may also assist in implementation of the air, vapour, and thermal control layers in highly energy-efficient building envelopes.

### 3.2. Mass Timber Envelopes

Hygroscopic properties of wood pose a great challenge to mass timber construction and require careful moisture management strategies, which should be planned in advanced and include passive and active solutions. To provide moisture protection during storage, transport, and operation, an effective passive strategy includes adhering weather-resistant membranes to the external surface of the CLT panel during manufacturing [39]. It is also critical to protect the exposed end-grain of the timbers, especially in vertical exposures where rain can run directly into the non-edge seams during the construction phase. Other passive strategies include installing capping joints during construction or sealing end-grains during manufacture with water resistant coatings. However, it should be noted that coatings are not completely effective, as moisture may slowly be absorbed into the wood. Just-in-time delivery and tenting protection may also contribute to moisture management during construction. While prevention remains the best solution, active moisture management strategies should be acted upon immediately during construction to remove any bulk liquid water found on the surface of the CLT panels.

Many of the envelope best-practice design principles in the literature for cold-climates internationally were found to be just as applicable to CLT assemblies; however, some strategies proved critical. Glass et al. [40] recommends an air-tight layer in all climates in order to reduce condensation caused by outside air coming into contact with indoor

conditions within the building envelope. Though CLT panels can be considered airtight under certain circumstances, it should not be relied on, especially if unprotected during construction and exposed to moisture [41]. The air control barrier should instead be continuously wrapped around all CLT components, especially joints, penetrations, and interfaces. They also advise that vapour barrier face-sealed building envelopes are not suitable for CLT construction in any climate. These type of façade systems, which employ vapour barriers without a ventilated cavity, are not recommended because they require perfect sealing of all penetrations and materials at the exterior face of the cladding, which is unrealistic in practice and has no redundancy to cater for failures of the control barrier. Allowing redundancy is vital to minimise the risk of moisture damage and ensuring protection against driving rain [42]. This is confirmed by Brambilla and Gasparri [43] in their Australian study, which recommends moisture-open strategies for timber-based envelopes.

Glass et al. [40] also recommends the use of drained and ventilated envelope systems, where cladding ventilation can be an important drying mechanism depending on the water storage capacity of the building cladding and the specific climate of the site. Staube and Finch [44] completed field and laboratory research showing that cladding ventilation has the potential to increase drying and reduce wetting from absorptive claddings and sun-driven moisture. Likewise, Wang and Ge [45] verified this fundamental approach through simulations and tests of CLT wall systems in the cold and temperate climate of Quebec City, Canada. They tested the sensitivity of drained and ventilated fibre cement cladding systems when subjected to different amounts of rain leakage. Their results indicate that drained and ventilated cavities successfully remove water and can return the CLT to an acceptable moisture content. Their results also indicate that low vapour permeability barriers are more successful in maintaining low moisture levels in the CLT if façade leaks are not present. When façade leaks are present, high vapour permeability barriers are preferred as they enable faster drying.

While there are a number of best practice guides for mass timber envelopes in cold and temperate climates [46,47], guidance for hot and humid climates is lacking, specifically for the Australian context. This is significant as CLT PH buildings in cold and temperate climates have been designed to deal with heat retention, cavity condensation and moisture removal during long, cold, wet winters, while they face different conditions in tropical climates: cooling regimes in response to high relative humidity and temperatures dominate their energy consumption, exerting a vapour pressure towards the interior of the building for much of the year—which is opposite to the vapour pressure direction in heating-dominated climates. A 'perfect wall' system for mass timber in these climates has yet to be confirmed.

There are additional risks when employing mass timber in tropical climates, these including the occurrence of termites and beetles [48]. Preservative treatments can be considered to improve wood resistance by employing impregnation of chemicals that are toxic to these organisms [36,40]. However, treatments should not be mistaken as a moisture control measure, as they do not prevent moisture ingress. Moisture issues may instead occur on other biologically sensitive building assembly materials. It is also worth noting that when timber is treated with copper chrome arsenate (CCA), only galvanised metals can be employed for CLT connections or fixtures, as studies show that treated timber is more corrosive than untreated timber [49].

These heightened moisture risks for mass timber buildings in hot and humid climates require deeper investigations to ensure durability can be maintained for the lifetime of the building.

## 4. Methodology

The parametric study presented in this paper aims at understanding the influence of material properties, locations, and disposition regarding mould growth, to identify risk-free wall assembly and to derive design guidelines for highly energy-efficient mass

timber buildings in hot and humid climates. The study particularly focuses on understanding the wall performance with respect to insulation layer position and weather resistive barrier (WRB) vapour permeability rates. The risk of mould growth is assessed by the moisture design tool, WUFI Pro (Wärme und Feuchte Instationär) to understand the risk of biodeterioration for a mass timber wall under the climate conditions of subtropical and tropical Australian climates. This tool is selected as it allows for the realistic calculation of dynamic coupled heat and moisture transport in the building envelope [50].

The parametric study is performed to assess the capacity for the selected assemblies to control the growth of mould by depriving fungi of moisture. The material parameters assessed are categorised to allow a comparison of results as per Figure 1. The material parameter include:

- Three insulation locations: external insulation, internal insulation, and split insulation.
- Three cases WRB locations: Case 1 has no WRB, Case 2 has a WRB adhered to the exterior of the CLT panel, and Case 3 has the WRB located on the exterior of the insulation layer.
- Three types of insulation products have been assessed: mineral wool, EPS, and wood fibre.
- Three classes of WRB have been assessed with different vapour permeance: Classes 2, 3, and 4.

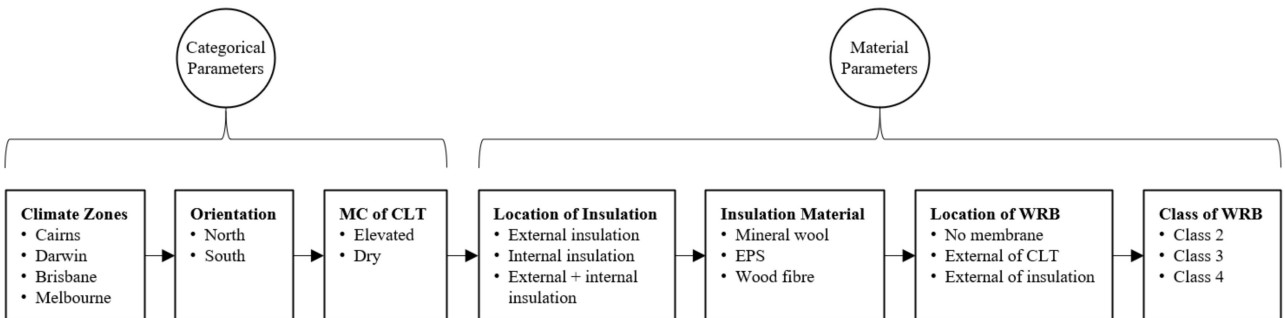

**Figure 1.** Parameters of interest iterated one at a time through the hygrothermal simulations.

The next sections explain in detail the rationale for the case-study employed, and the assumptions and inputs.

### 4.1. Identification of a Suitable Case-Study

This study employs Gillies Hall, the largest mass timber PH building in the southern hemisphere (Figure 2), as a case-study for verifying the hygrothermal behaviour of an energy efficient and durable multi-storey mass timber building located in Australian hot and humid climates. Gillies Hall is one of Monash University's student residential buildings, which is located near Melbourne, in a temperate oceanic climate. The case-study design is hypothetically transferred to three Australian hot and humid climates. Each climate-specific iteration of the pilot project is re-designed to achieve compliance with the PH standard—while also complying with the minimum energy efficiency provisions of the Australian NCC 2019 in Section J Volume 1. To ensure the durability of the re-designed pilot projects, hygrothermal analysis is used to identify those CLT wall assembly solutions that effectively minimise the risk of mould growth and corrosion.

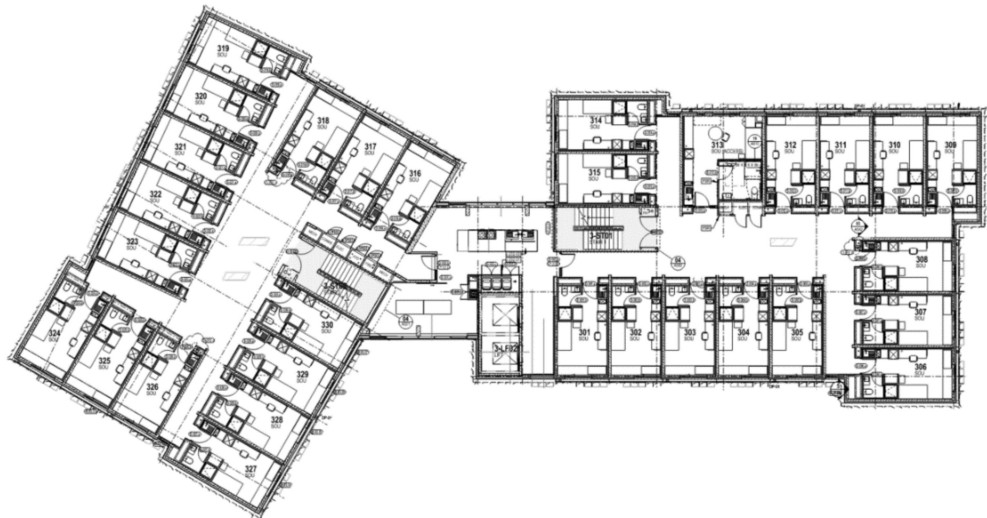

**Figure 2.** Floor plan of the case study building, Gillies Hall (retrieved from Jackson Clements Burrows Architects).

### 4.2. Input Data and Assumptions

The reliability of the hygrothermal simulations using the methodology outlined above is directly related to the input parameters. The input parameters are organised under the following sections: outdoor and indoor climate, material properties, boundary conditions, and moisture safety assessment criteria.

### 4.2.1. Outdoor Climate and Orientation

For this study, the pilot project building, originally designed for a temperate oceanic climate, is simulated in the climates listed in Table 1: Melbourne, Brisbane, Cairns, and Darwin. These cities represent the main Australian tropical and subtropical climates, except for Melbourne, which is the location of the constructed pilot project and is included as the reference case for a temperate climate.

**Table 1.** Köppen climate zones [51] considered in the paper. Climate: A (tropical), m (monsoon), w (savanna), C (temperate), f (fully humid), a (hot summer), b (warm summer).

| Climate | Description | City |
|---|---|---|
| Am | Tropical monsoon climate | Cairns |
| Aw | Tropical savanna climate | Darwin |
| Cfa | Humid subtropical climates | Brisbane |
| Cfb | Temperate oceanic climate | Melbourne |

The simulations are performed for the north and south elevations. The north orientation was selected to consider solar-driven inward diffusion, which may be observed when the sun shines on damp cladding, while the south orientation has minimal solar drying effects and, consequently, a reduced drying process.

The climatic files used for the hygrothermal simulations are in accordance with ASHRAE 160-2016 for the use of 10 consecutive years of measured weather data. These weather data are a superior representation of severe conditions in comparison to typical weather datasets. Typical weather datasets such as typical mereological year (TMY) are synthesized to represent climatic long-term trends, which are favoured for energy-use simulations [52]. However, due to the accessibility of TMY weather files and the current inaccessibility of cleaned and processed measured weather data within Australia, TMY weather files are generally used for hygrothermal simulations in the building industry. The use of a different climatic files in this study is significant, as it may lead to more critical results. This choice is supported by ASHRAE research project RP-1325 [53], which

notes that the climatic dataset should impose a greater level of stress than the average climate. In contrast with a typical weather year, the use of 10 consecutive years of measured weather data provides a level of safety in terms of moisture performance and durability, as a 10-year period would include extreme situations that are to be expected once every 10 years. The ASHRAE research project notes that climatic datasets have been developed that may impose an even greater level of stress, such as climate datasets using a moisture design reference year or a construction-dependent damage function weather set. These developments are outside of the scope of this paper, as there is no established consensus in Australia for these developing weather selection methods.

The 10-year period of measured data that is used in this study originates from the National Centres for Environmental Information (NCEI) archives. The recorded hourly dry bulb temperature, dew point temperature, precipitation, wind direction, wind speed, atmospheric pressure, and cloud index were taken from the site of the international airport at each respective location selected in this study. The data were then cleaned and processed, for use in WUFI. This was done by interpolating missing values, removing duplicates and using the Zhang Huang solar model [54] to approximate the diffuse, direct, and global horizontal solar irradiation. Average temperature and relative humidity, along with the monthly sum of precipitation for the 4 selected climates based on the processed NCEI archive, are compared in Figure 3.

The uncertainty of rainfall data in this study should also be noted, which is highly relevant as the effect of rainfall data on hygrothermal performance can have a significant effect on simulated mould growth. Cornick et al. [55] found that there was uncertainty associated with the use of 6 h precipitation times steps, as utilised by the NCEI archive. It has been demonstrated that measuring rainfall data in hourly timesteps may not even be granular enough and may still produce errors in wind-driven rain [56].

The year ranges used in this study were chosen based on the most recent 10-year range where less than 5% of annual rainfall data is missing from the NCEI archives. The selected Melbourne year range was 2005:2014, while Brisbane, Cairns, and Darwin were all based off the year range 2011:2020. The 6 h precipitation values were then cleaned and distributed across the 6 h measurement period. The hygrothermal simulations were then run for the entire constructed 10-year weather file to account for long-term hygrothermal phenomena.

### 4.2.2. Indoor Conditions

The indoor climate temperature is defined according to the PH standard. The standard allows an internal temperature range of 20 to 25 °C, which can be exceeded for up to 10% of the year, while the indoor relative humidity may exceed 60% for a maximum 20% of the year. These conditions are simulated in WUFI with a temperature set-points of 20 to 26 °C. Internal conditions are defined according to the ASHRAE 160 intermediate method, which couples the internal and external temperature and RH. The indoor RH is also influenced by the interior humidity generation and the humidity-controlled dehumidification equipment that maintains the interior RH below 70%. Limitation in modelling the interior conditions within WUFI mean that the PH standard threshold of 60% RH will be exceeded for much longer than the 20% limit. The outcome of this limitation will lead to conservative results for a building operated as per PH standard requirements, as the interior relative humidity in the hygrothermal simulation will be higher than in a Passivhaus-certified building.

The indoor climate (Table 2) is based on a single-bedroom studio apartment in the pilot project with a typical volume of 114 m³. This zone is modelled with air exchange rate of 0.2 ACH. This is to represent the minimum air supply of 30 m³/h per person for hygienic air supply rate as per EN 1946 Part 6. This expected to be delivered by the continuous mechanical ventilation system. A WUFI default moisture generation rate of $8.06 \times 10^{-5}$ kg/s is applied to the zone.

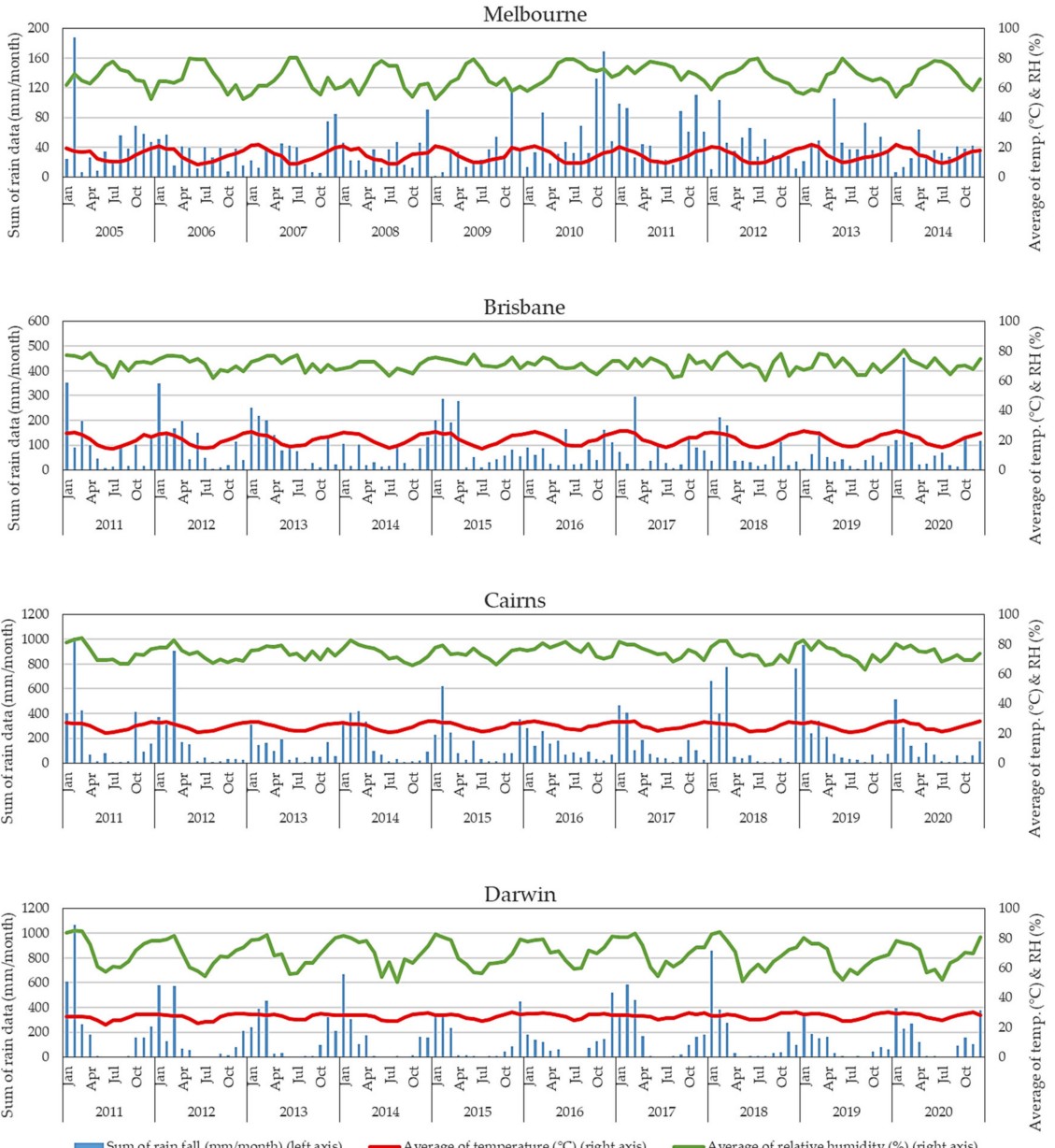

**Figure 3.** 10-year climate files generated from the NCEI archives. Note the different range of years used for Melbourne and the different y-axis bounds for the sum of monthly rain data.

**Table 2.** Indoor conditions.

| Condition | Setting |
|-----------|---------|
| System | AC with dehumidification |
| Temperature | Heating set-point of 20 °C<br>Cooling set-point of 26 °C |
| Relative humidity | Relative humidity limit of 70% |
| Air exchange rate | 0.2 air changes per hour |
| Moisture generation | $8.06 \times 10^{-5}$ kg/s |

### 4.2.3. Envelope Performance

The authors previous study [57] re-designed the envelope of the same pilot project used in this paper to achieve the PH standard in hot and humid Australian climates. These characteristics for Passivhaus compliance informed the thermal performance of the wall

assemblies used in this paper for the selected climates. The thermal performance required to maintain compliance with the Passivhaus standard in the selected climates is shown in Table 6. The pilot project wall assembly is depicted in Figure 4, while the order of layers in the wall assembly are altered in the subsequent modelling to understand if different assembly variations result in superior hygrothermal performance. The wall assembly variations are highlighted in the following sections.

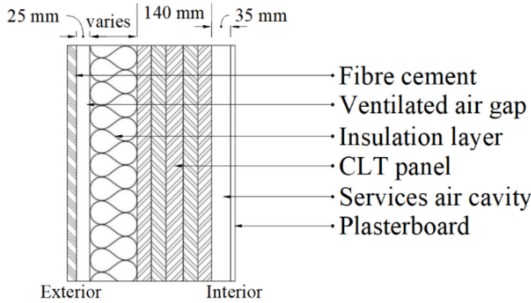

**Figure 4.** Cross-section of materials used in wall assembly.

### 4.2.4. Material Properties

All material properties are based on WUFI's in-built material database, where available. These properties have been experimentally quantified and validated by the Fraunhofer Institute of Building Physics (Table 4). This study considers three variations to the CLT wall assembly, where insulation is either fixed to the exterior of the CLT surface, installed to the interior, or split and installed to both the internal and external surface of the CLT panel (Figure 5). The performance for a range of vapour permeable WRBs to the external of the CLT structure have been evaluated in this study. Three high quality polypropylene and monolithic thermoplastic elastomer ether ester (TEEE) vapour control membrane products have been analysed, available off-the-shelf to the Australian market. The vapour permeability classes and equivalent meters of still air ($S_D$ value) have been based on the membrane supplier datasheets and inputted into WUFI (Table 3). The vapour permeability of Class 2 can be defined as a vapour barrier, while Class 3 and 4 is defined as vapour permeable according to Australian Standard 4200 [58]. It should also be noted that the Class 2 membrane is not vapour impermeable, as it allows some vapour diffusion.

The other materials composing the CLT assembly including fibre cement, CLT panel, insulation layer, and plasterboard are given the hygrothermal properties in Tables 4–6 and depicted by Figure 4. The WRB position within the assembly is one of the variables tested in this study. While it is used as the primary rain barrier during construction, the WRB acts as the secondary wind and air barrier during operation, preventing air convection and moisture ingress, in the form of air infiltration and driving rain into the insulation layer and the CLT structure [59]. Figure 5 depicts the various locations where the WRB might be installed within the assembly and thus simulated to identify how WRB placement affects moisture control. Case 1 is the reference case, where no WRB is included in the wall assembly. In Case 2, the WRB is adhered to the exterior of the CLT panel. Lastly, Case 3 has the WRB located on the exterior of the insulation layer. It should be noted that the internal insulation variation has only two cases, as the wall assembly should always maintain a ventilated air cavity internal to the cladding, as explained in Section 3 of this paper.

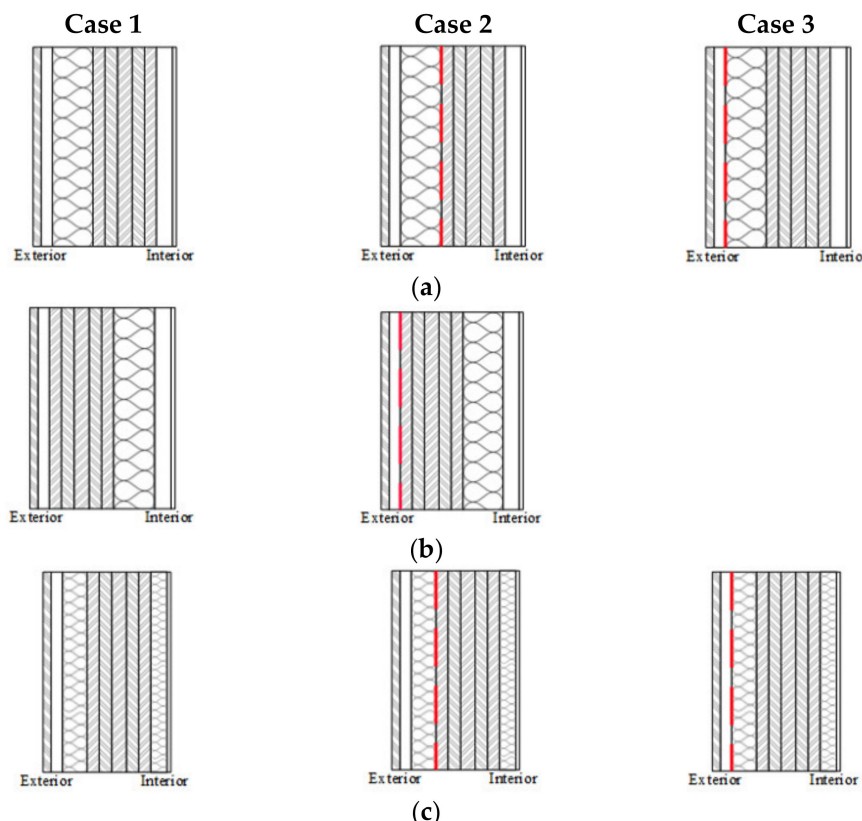

**Figure 5.** Wall assembly variations with external insulation (**a**), internal (**b**), and split (**c**). No WRB modelled (Case 1), WRB adhered to exterior of CLT panel (Case 2), and WRB installed to exterior of external insulation layer (Case 3).

**Table 3.** Vapour control membrane classification as per Table 4 in AS 4200.1:2017, Pliable building membranes and underlays. $S_D$ values for the Class 3 and 4 membranes are in accordance with ASTM-E96 Method B wet cup—23 °C at 50% RH, while $S_D$ values for the Class 2 membranes are in accordance with EN 1931 wet cup—23 °C at 75% RH.

| Classification | Vapour Permeance (µg/Ns) | Equivalent Meters of Still Air, $S_D$ Value, (m) |
|---|---|---|
| Class 2 | 0.09 | 2.25 |
| Class 3 | 0.4 | 0.5 |
| Class 4 | 2.17 | 0.09 |

**Table 4.** Hygrothermal properties of the materials used in the multi-layered assemblies within the WUFI simulations.

| Material | Thickness (m) | Bulk Density (kg/m$^3$) | Porosity (m$^3$/m$^3$) | Specific Heat Capacity (J/kg K) | Vapour Diffusion Resistance Factor (-) | Thermal Conductivity (W/m·K) |
|---|---|---|---|---|---|---|
| Fibre cement | 0.008 | 1610 | 0.16 | 850 | 83.3 | 0.13 |
| Ventilated air gap | 0.025 | 1.3 | 0.999 | 1000 | 0.56 | 0.13 |
| CLT panel | 0.140 | 500 | 0.858 | 1880 | 1734.4 | 0.119 |
| Services air cavity | 0.035 | 1.3 | 0.999 | 1000 | 0.56 | 0.13 |
| Plasterboard | 0.015 | 850 | 0.65 | 850 | 8.3 | 0.2 |

Variation of insulation layer properties discussed in the subsequent section.

**Table 5.** Hygrothermal properties of insulation material used in the WUFI simulation.

| Material | Bulk Density (kg/m$^3$) | Porosity (m$^2$/m$^3$) | Specific Heat Capacity (J/kg K) | Water Vapour Diffusion Factor (-) | Thermal Conductivity (W/m·K) |
|---|---|---|---|---|---|
| Mineral wool | 178 | 0.934 | 850 | 1.76 | 0.0336 |
| EPS | 14.8 | 0.99 | 1470 | 73.01 | 0.036 |
| Wood fibre | 155 | 0.981 | 1400 | 3 | 0.042 |

**Table 6.** Wall R-value (given by WUFI) of the Passivhaus pilot projects in the selected loca-tions, exclusive of 2D and thermal bridging effects. The insulation thickness for each location is also given.

| Climate | Melbourne | Brisbane | Cairns | Darwin |
|---|---|---|---|---|
| R-value (m$^2$ K/W) | 4.45 | 3.56 | 3.56 | 3.86 |
| Mineral Wool (mm) | 90 | 60 | 60 | 70 |
| EPS (mm) | 95 | 63 | 63 | 74 |
| Wood Fibre (mm) | 111 | 74 | 74 | 86 |

The thermal performance of the wall assembly for the Pilot Project as defined in the paper [57] is shown in Table 6. The insulation thicknesses for the selected insulation products are dimensioned to the minimum possible to maintain compliance with both the PH standard and Section J of the NCC for each climate. Three different types of insulation materials are selected for this parametric study: a mineral, an organic and a synthetic material. The mineral insulation products (known as mineral wool or rock wool) commonly used in commercial construction applications are manufactured from basalt and comply with fire requirements in Australia. Therefore, a material with similar properties was selected from the WUFI in-built material database: *ROXUL Comfortboard 110* product. The organic insulation material selected was wood fibre as it has found greater adoption internationally and has started to enter the Australian market, where some of these products are impregnated with fire retardants. The *wood fibre insulation board* product was selected from the WUFI database for this category. The synthetic insulation material selected was expanded polystyrene (EPS) insulation, which was chosen as it is lightweight, low cost, and available to the Australian market as an insulted fire-retardant panel. The EPS material selected in WUFI was the *expanded polystyrene insulation* product. The hygrothermal properties for the three insulation materials are given in Table 5.

The case study's CLT panels were sourced from a European spruce CLT supplier. However, Australian suppliers of radiata pine CLT are increasing their supply to the Australian market, thus, to ensure the relevance of this study for the Australia context, the southern yellow pine (North American database [60]) was employed in the simulation as it was the closest timber species to radiata pine available in the WUFI material library.

The cladding is modelled as fibre cement panels and acts principally as a rain control and ultraviolet screen. The 25 mm drained and ventilated air cavity is modelled with 10 ACH. Simpson [61] calculated that fibre cement cladding may have a mean ACH of 89 with a standard deviation of 36 for continuous slot vents of 1 mm thickness at top and 12 mm at bottom. Therefore, 10 ACH represents a conservative ACH value, which statistically 98% of projects would exceed based on Simpson's findings. The reasoning for selecting a conservative (low) ACH is that the results of this study can be applicable to a larger number of projects with unknown flashing and vented air cavity designs and therefore greater drying potentials. A high ACH is preferred for moisture-safe design, as it increases the capacity to vent and remove moisture that would otherwise build-up in the air cavity space. It is worth noting here that, for all climate zones considered in this study, the thermal performance of the wall assemblies achieves Australian code-compliance for energy-efficiency, while other façade requirements, such as fire and acoustic performance, are not considered within the scope of this study.

### 4.2.5. Surface Boundaries and Initial Conditions

The study employs two different initial conditions. The first initial condition represents building assemblies that are protected from rain events, where a membrane has been adhered to the CLT during construction, and any exposed end-grains are protected from stormwater. In this case, where no wetting of the CLT panels occurs during construction, the initial equilibrium water content of the assembly materials has been set to a relatively low initial RH of 80%. This is equivalent to a moisture content of approximately 12%. A summary of the results for the protected CLT panels are in Appendix A.

The second initial condition is that the CLT panels are unprotected and exposed to stormwater during the construction phase. This leads to an elevated level of moisture content within the CLT layer. The recommendation in ASHRAE 160-2016 for assemblies unprotected during construction is to multiply the equilibrium water content at a relative humidity of 80% of the material layers by a factor of two, which has been implemented in these cases. This is equivalent to a moisture content of approximately 24%. This represents moisture ingress from rain events that had occurred during construction where CLT panels are left unprotected. The elevated moisture content is modelled as being absorbed uniformly through the CLT profile, which may happen at geometric junctions or at a window rough opening where the end-grain is exposed. A summary of the results for the unprotected CLT panels are in Appendix B.

ASHRAE 160 guidance states that a small amount of wind driven rain (1%) will penetrate behind the cladding, even if adequate flashing is included in the design, based on the work by Lacy [62]. This is simulated as being deposited on the material directly facing the ventilated air cavity where applicable; either in the insulation or CLT layer, or on the surface of the WRB. Boundary conditions, shown in Table 7, are applied as per a typical multi-storey building clad with fibre cement.

**Table 7.** Boundary conditions of the internal and exterior surfaces.

| Parameter | Exterior Surface | Interior Surface |
|---|---|---|
| Heat transfer coefficient (W/m$^2$·K) | Wind dependent | 8 |
| Short-wave radiation emissivity | 0.6 | - |
| Long-wave radiation emissivity | 0.9 | - |
| Rain exposure factor | 1.2 | - |
| Rain deposition factor | 0.5 | - |

### 4.3. Moisture-Safety Criteria

ASHRAE 160-2016 uses mould growth on material substrate (A) and corrosion rate (B) as indicators to predict potential moisture safety or risk. Exceeding either criterion is considered a failure event.

Technical Research Centre of Finland (VTT) have developed a realistic mould growth index (MGI) which is based on the growth of different mould fungi on the surface of pine sapwood in different conditions, including the effects of temperature, relative humidity, and mould exposure time [63]. The MGI defines the moisture safety criteria (A) used in this study. Complying with this criterion requires that the CLT surface does not exceeding an MGI of 1, this corresponds to mould growth not yet germinating. Further investigation is defined for a MGI greater than 1 to 3, which corresponds to mould growth only visible under a microscope. While this range of mould growth complies with ASHRAE 160, additional project-specific hygrothermal simulations may be needed for assessing acceptability in applying the solution to specific projects. Results that exceed a MGI of 3 are indicated with a failure event.

The MGI requires that a mould growth sensitivity class is assigned to the material under analysis. These classes adjusts different mould growth phenomena to correlate with empirical mould growth studies on the corresponding materials [64]. The building material surface under analysis, the CLT panel, is assessed as a 'sensitive' material class

with 'almost no mould decline,' corresponding to a pine-sapwood wood-based board. Treated CLT products may have lower sensitivity, but their analysis falls outside the scope of this assessment.

The second moisture safety criteria are defined by the corrosion rate (B) of metallic fasteners embedded in the CLT panel. The prevention of corrosion derives from the properties and function of the metals used in the assembly. The corrosion criterion of ASHRAE 160 describes a critical relative humidity at which the corrosion rate become rapid. Harriman et al. [65] shows that clean iron will not corrode until the air is almost saturated, while different metal alloys have critical RH ranging from 70% to 87%. If no information on the metal is available, the corrosion rate indicator (B) for moisture safety is defined as follows: the 30-day running average surface RH not exceeding 80% is indicated with a pass to prevent corrosion on a metallic fastener or connector. Exceeding this 30-day running average surface RH leads to a failure event for the corrosion rate criterion.

## 5. Results and Discussion

The following sections evaluate the CLT wall assembly solutions against the moisture-safe criteria as defined by ASHRAE 160, specifically in respect to the position and type of the insulation layer and WRB.

The numerical quality of the simulation results is checked by recording the number of convergence failures and the difference in moisture balance over the duration of the 10-year simulation. One convergence failure represents a single hour time step within the simulation where the iterative calculation process does not converge on a numerical solution within the maximum allowed number of iterations. If this is the case, WUFI accepts the last iteration, records a convergence failure, and then continues to the next time step. However, a convergence failure may not indicate an error. Fifty convergence failures per year may indicate a problem, but not in all cases [66]. This is because the iterative calculation may have been very close to converging on a final solution before reaching the maximum number of iterations. For this reason, the difference in moisture balance at the left and the right surface of the assembly was also recorded. Ideally, this difference should be zero since the long-term change in the total water content results from moisture being transported through the surfaces should be identical. However, convergence failures can cause discrepancies.

WUFI distributes the modelled assembly across grid elements. The monitoring positions of all subsequent results were placed in both the inner-most grid (internal surface) and the outer-most grid (external surface) of the CLT panel. The grid distribution allows the heat and moisture transport equations to be implement for each grid element. To limit the convergence failures and difference in moisture balance, the numerical grid in WUFI was generated as the fine grid resolution setting. This ensures that the temperature and moisture distributions across the assembly can be appropriately resolved, avoiding kinks in the temperature and moisture profiles due to poor resolution. Generally, the convergence failures in the results were very low, but greater than 100 in some instances. In those cases, the difference in moisture balance was close to zero; therefore, no further action was taken to improve the numerical quality of the simulations.

### 5.1. External Insulation Cases

The first analysis focuses on the external CLT solutions, as depicted in Figure 6. The performance criteria result for protected assemblies are presented in Table 8. Full results can be found in Appendices A and B.

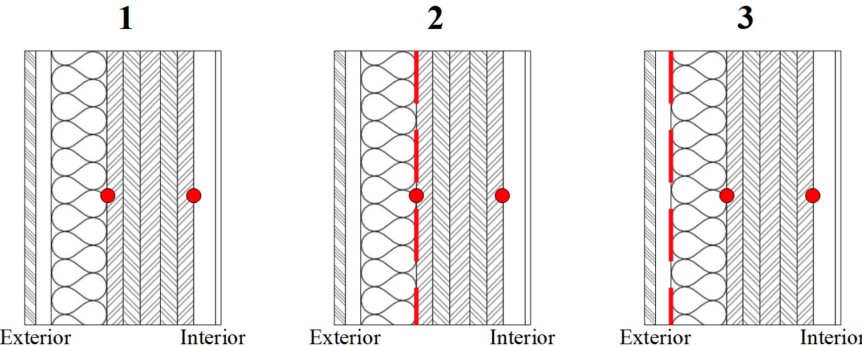

**Figure 6.** Wall assembly with exterior insulation. No WRB modelled (Case 1), WRB adhered to exterior of CLT panel (Case 2), and WRB installed to exterior of insulation layer (Case 3). Red points represent the monitoring positions at the internal and external surface of the CLT panel.

**Table 8.** Protected CLT results for external insulation. O (orientation). Performance criteria: A (mould growth index), B (corrosion rate). Classification of WRB: Class 2, 3, and 4 (vapour permeance). Green (pass), red (fail), or yellow (further investigation).

| Protected Case 1 | | Mineral Wool | | | | EPS | | | | Wood Fibre | |
|---|---|---|---|---|---|---|---|---|---|---|---|
| | | No WRB | | | | No WRB | | | | No WRB | |
| Climate | O | A | B | | | A | B | | | A | B |
| Cairns | N | 0.78 | 3.9 | | | 0.15 | 1.9 | | | 0.65 | 3.8 |
| | S | 3.76 | 42.9 | | | 0.47 | 3.1 | | | 3.56 | 43.6 |
| Darwin | N | 2.73 | 29.9 | | | 0.55 | 2.7 | | | 3.24 | 24.6 |
| | S | 1.45 | 22.5 | | | 0.3 | 2.7 | | | 1.24 | 12.7 |
| Brisbane | N | 0.14 | 0 | | | 0.03 | 0 | | | 0.07 | 0 |
| | S | 0.31 | 1.5 | | | 0.07 | 0.3 | | | 0.2 | 0.6 |
| Melbourne | N | 0 | 0 | | | 0 | 0 | | | 0.01 | 0 |
| | S | 0.06 | 0 | | | 0 | 0 | | | 0 | 0 |

| Protected Case 2 | | Mineral Wool | | | | | | EPS | | | | | | Wood Fibre | | | | | |
|---|---|---|---|---|---|---|---|---|---|---|---|---|---|---|---|---|---|---|---|
| | | Class 2 | | Class 3 | | Class 4 | | Class 2 | | Class 3 | | Class 4 | | Class 2 | | Class 3 | | Class 4 | |
| Climate | O | A | B | A | B | A | B | A | B | A | B | A | B | A | B | A | B | A | B |
| Cairns | N | 0.06 | 1.1 | 0.48 | 2.7 | 0.68 | 3.8 | 0.15 | 1.9 | 0.12 | 1.8 | 0.15 | 1.9 | 0.61 | 3.6 | 0.47 | 2.9 | 0.61 | 3.6 |
| | S | 0.66 | 8.9 | 2.2 | 34.4 | 3.39 | 41.5 | 0.27 | 2.9 | 0.41 | 3.1 | 0.46 | 3.1 | 0.71 | 14.4 | 2.25 | 35.5 | 3.27 | 42.5 |
| Darwin | N | 0.75 | 3 | 1.54 | 14.6 | 2.16 | 28.3 | 0.35 | 2.6 | 0.49 | 2.7 | 0.54 | 2.7 | 1 | 4.1 | 1.99 | 11.6 | 2.82 | 20.3 |
| | S | 0.46 | 2.9 | 1.02 | 9.2 | 1.34 | 18.2 | 0.1 | 1.6 | 0.25 | 2.6 | 0.29 | 2.7 | 0.46 | 3.3 | 0.91 | 3.4 | 1.16 | 11.1 |
| Brisbane | N | 0.03 | 0 | 0.05 | 0 | 0.08 | 0 | 0.01 | 0 | 0.03 | 0 | 0.03 | 0 | 0.03 | 0 | 0.06 | 0.1 | 0.07 | 0.1 |
| | S | 0.08 | 0.4 | 0.14 | 0.4 | 0.23 | 1 | 0.04 | 0.1 | 0.06 | 0.3 | 0.07 | 0.3 | 0.09 | 0.6 | 0.16 | 0.7 | 0.19 | 0.6 |
| Melbourne | N | 0 | 0 | 0 | 0 | 0 | 0 | 0 | 0 | 0 | 0 | 0 | 0 | 0 | 0 | 0.01 | 0 | 0.01 | 0 |
| | S | 0 | 0 | 0.01 | 0 | 0.04 | 0 | 0 | 0 | 0 | 0 | 0 | 0 | 0 | 0 | 0 | 0 | 0 | 0 |

| Protected Case 3 | | Mineral Wool | | | | | | EPS | | | | | | Wood Fibre | | | | | |
|---|---|---|---|---|---|---|---|---|---|---|---|---|---|---|---|---|---|---|---|
| | | Class 2 | | Class 3 | | Class 4 | | Class 2 | | Class 3 | | Class 4 | | Class 2 | | Class 3 | | Class 4 | |
| Climate | O | A | B | A | B | A | B | A | B | A | B | A | B | A | B | A | B | A | B |
| Cairns | N | 0.18 | 2 | 0.47 | 2.2 | 0.65 | 3.6 | 0.05 | 0.9 | 0.17 | 1.9 | 0.2 | 2 | 0.31 | 2.8 | 0.47 | 2.5 | 0.59 | 3.2 |
| | S | 0.56 | 4.9 | 1.52 | 32.4 | 2.95 | 40 | 0.34 | 3.1 | 0.51 | 3.6 | 0.58 | 5.1 | 0.53 | 3.4 | 1.04 | 26.3 | 2.1 | 37.3 |
| Darwin | N | 0.73 | 2.8 | 1.48 | 14.9 | 2.01 | 28.9 | 0.4 | 2.6 | 0.56 | 2.8 | 0.62 | 2.8 | 0.82 | 3.6 | 1.33 | 8.3 | 1.72 | 19.4 |
| | S | 0.4 | 2.8 | 0.93 | 10.1 | 1.22 | 20.8 | 0.17 | 2.4 | 0.3 | 2.7 | 0.35 | 2.8 | 0.43 | 2.8 | 0.79 | 3 | 1.04 | 10.9 |
| Brisbane | N | 0.03 | 0 | 0.04 | 0 | 0.06 | 0 | 0.01 | 0 | 0.03 | 0 | 0.03 | 0 | 0.07 | 0.3 | 0.07 | 0.2 | 0.07 | 0.1 |
| | S | 0.12 | 0.4 | 0.2 | 0.7 | 0.31 | 1.3 | 0.06 | 0.3 | 0.08 | 0.4 | 0.08 | 0.4 | 0.13 | 0.8 | 0.17 | 0.8 | 0.21 | 0.7 |
| Melbourne | N | 0.01 | 0 | 0 | 0 | 0 | 0 | 0 | 0 | 0 | 0 | 0 | 0 | 0.06 | 0.3 | 0.02 | 0 | 0.02 | 0 |
| | S | 0 | 0 | 0 | 0 | 0.01 | 0 | 0 | 0 | 0 | 0 | 0 | 0 | 0.02 | 0 | 0.01 | 0 | 0 | 0 |

The results in Table 8 show that all solutions simulated are unlikely to produce mould growth failure events, except for the tropical climate cases. This is predominately due to CLT being a robust solution when compared to stick frame timber construction [43]. Interior conditions in a PH certified building also reduce the likelihood of mould growth. This is confirmed by Langer et al. [67], which found lower interior RH and microbial activity for assessed PH buildings than for comparative conventional buildings. Case 1 and 2 shows a reduction in risk of MGI and corrosion for all assemblies, due to the adhered WRBs providing protection from driving rain that splashes past the cladding.

These results show that the corrosion criterion is more stringent than the mould growth index criterion for all scenarios. This is confirmed by other studies in the literature [68,69]. The corrosion rate failures in tropical climates are primarily caused by high ambient vapour pressure throughout the year. This shows that corrosion protection is critical in tropical

climates for metal fasteners and connections. Corrosion risk is also shown to be higher for the south orientation scenario, which has a lower solar load and resultingly lower drying potential. However, it should be noted that the reduced drying potential effect of the solar load appears to be eclipsed by the effect of the prevailing rain direction, as this controls the deposition of the driving rain on the WRB. This can be seen in the Darwin results, which show a greater number of corrosion failure events and higher MGIs for the north orientation, with prevailing rain, compared to the south orientation.

Regarding the influence of the WRB classes, the lowest MGI and corrosion rates occur for the temperate and subtropical climates with the Class 4 WRB, while the lowest MGI and corrosion rates occur for the tropical climates with the Class 2 WRB. This is because the WRB with a higher vapour resistance reduces the moisture that diffuses through the assembly in hot and humid climates. It can also be seen that locating the WRB on the external of the insulation layer, as modelled in Case 3, slightly reduces the risk of mould growth.

Considering the influence of the insulation materials, wood fibre has an advantage over mineral wool for external insulation because it has twice the vapour resistance. For a similar reason, EPS is a robust solution where the CLT panels are kept dry during building use, as it functions as a vapour control layer during operation.

However, as shown in Appendix B or Table 9 below for Case 2, when the CLT assemblies are unprotected during construction and the moisture content (MC) of the CLT panels is elevated, the EPS reduces drying potential. This is because the rigid insulation traps moisture into the assembly. This case leads to a poorer outcome with more failure events, where the elevated moisture content is unable to dry. For the mineral wool and wood fibre assemblies that are left unprotected during construction, only the temperate and subtropical climates with low vapour resistant WRBs have a low risk for mould growth. In instances where storm-water management strategies have failed, heaters or fans could be considered internally to reduce the MC, though this solution has not been tested in this study.

**Table 9.** Unprotected CLT results for external insulation. O (orientation). Performance criteria: A (mould growth index), B (corrosion rate). Classification of WRB: Class 2, 3, and 4 (vapour permeance). Green (pass), red (fail), or yellow (further investigation).

| Unprotected Case 2 Climate | O | Mineral Wool Class 2 | | Class 3 | | Class 4 | | EPS Class 2 | | Class 3 | | Class 4 | | Wood Fibre Class 2 | | Class 3 | | Class 4 | |
|---|---|---|---|---|---|---|---|---|---|---|---|---|---|---|---|---|---|---|---|
| | | A | B | A | B | A | B | A | B | A | B | A | B | A | B | A | B | A | B |
| Cairns | N | 2.57 | 15.5 | 1.9 | 7.3 | 1.78 | 6.2 | 2.32 | 14 | 2.38 | 14.3 | 2.32 | 14 | 1.8 | 6.2 | 1.97 | 7.9 | 1.8 | 6.2 |
| | S | 2.99 | 31.7 | 2.98 | 43.8 | 3.39 | 47.7 | 3.05 | 26 | 2.96 | 25.3 | 2.92 | 25.6 | 3.02 | 35.6 | 2.97 | 45.4 | 3.29 | 48.3 |
| Darwin | N | 2.63 | 13.4 | 2.67 | 19.6 | 3.03 | 30.6 | 2.62 | 14.3 | 2.59 | 13.7 | 2.59 | 13.6 | 2.79 | 13.8 | 3.15 | 17 | 3.8 | 24 |
| | S | 2.67 | 14.9 | 2.38 | 17.7 | 2.36 | 22.8 | 2.9 | 18.6 | 2.79 | 17.5 | 2.75 | 17 | 2.72 | 15.2 | 2.38 | 13.4 | 2.29 | 16.7 |
| Brisbane | N | 2.02 | 13.4 | 1.32 | 3.1 | 1 | 2 | 2.55 | 21.6 | 2.29 | 15.3 | 2.18 | 14.4 | 2.08 | 13.7 | 1.47 | 3.6 | 1.12 | 2.9 |
| | S | 2.85 | 25.6 | 1.98 | 15.1 | 1.57 | 6.9 | 3.14 | 29.7 | 2.99 | 27.5 | 2.93 | 26.6 | 2.9 | 26.2 | 2.08 | 15.8 | 1.65 | 7.8 |
| Melbourne | N | 2.05 | 15.8 | 1.52 | 9.8 | 0.32 | 0.7 | 2.81 | 25.6 | 2.57 | 19.9 | 2.44 | 18.8 | 2.25 | 17 | 1.13 | 3.1 | 0.68 | 1.9 |
| | S | 2.69 | 20.1 | 1.55 | 9.9 | 0.65 | 1.6 | 3.28 | 38.6 | 3.01 | 29.3 | 2.92 | 27.5 | 2.84 | 21.9 | 1.46 | 8.9 | 0.99 | 2.7 |

### 5.2. Split Insulation Cases

The second analysis is for a split insulation solution, where insulation is located both internally and externally to the CLT panel, as depicted in Figure 7. The performance criteria result for protected assemblies are presented in Table 10. Full results can be found in Appendices A and B.

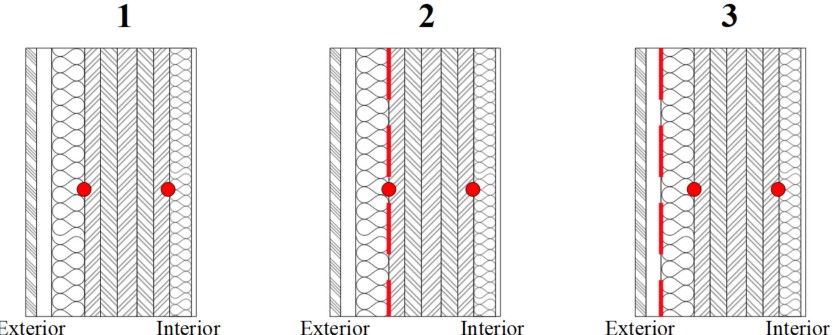

**Figure 7.** Wall assembly with insulation located both externally and internally. No WRB modelled (Case 1), WRB adhered to exterior of CLT panel (Case 2), and WRB installed to exterior of external insulation layer (Case 3). Red points represent the monitoring positions at the internal and external surface of the CLT panel.

**Table 10.** Protected CLT results for split insulation. O (orientation). Performance criteria: A (mould growth index), B (corrosion rate). Classification of WRB: Class 2, 3, and 4 (vapour permeance). Green (pass), red (fail), or yellow (further investigation).

| Protected Case 1 | | Mineral Wool No WRB | | EPS No WRB | | Wood Fibre No WRB | |
|---|---|---|---|---|---|---|---|
| Climate | O | A | B | A | B | A | B |
| Cairns | N | 0.53 | 3 | 0.1 | 0.7 | 0.49 | 3.2 |
| | S | 3.86 | 35.6 | 0.55 | 6.9 | 3.98 | 38.3 |
| Darwin | N | 2.15 | 15.2 | 0.68 | 2.6 | 3.71 | 15.4 |
| | S | 1.15 | 8.1 | 0.27 | 2.5 | 1.11 | 7.2 |
| Brisbane | N | 0.18 | 0 | 0.03 | 0 | 0.08 | 0 |
| | S | 0.4 | 1.7 | 0.07 | 0.2 | 0.19 | 0.5 |
| Melbourne | N | 0 | 0 | 0 | 0 | 0 | 0 |
| | S | 0.27 | 20.2 | 0 | 0 | 0.01 | 0 |

| Protected Case 2 | | Mineral Wool Class 2 | | Class 3 | | Class 4 | | EPS Class 2 | | Class 3 | | Class 4 | | Wood Fibre Class 2 | | Class 3 | | Class 4 | |
|---|---|---|---|---|---|---|---|---|---|---|---|---|---|---|---|---|---|---|---|
| Climate | O | A | B | A | B | A | B | A | B | A | B | A | B | A | B | A | B | A | B |
| Cairns | N | 0.43 | 2.8 | 0.28 | 1.8 | 0.43 | 2.8 | 0.09 | 0.6 | 0.05 | 0 | 0.09 | 0.6 | 0.44 | 3 | 0.32 | 2 | 0.44 | 3 |
| | S | 0.5 | 3.9 | 1.65 | 27.5 | 3.28 | 34.6 | 0.22 | 2.4 | 0.44 | 3.1 | 0.53 | 5.8 | 0.61 | 10.4 | 2.14 | 31.4 | 3.46 | 37.3 |
| Darwin | N | 0.64 | 2.7 | 1.39 | 7.1 | 1.91 | 13.8 | 0.35 | 2.4 | 0.57 | 2.6 | 0.66 | 2.6 | 0.91 | 3.5 | 1.87 | 8.2 | 2.93 | 14.2 |
| | S | 0.27 | 2.1 | 0.71 | 2.6 | 1.05 | 6.6 | 0.02 | 0 | 0.2 | 2.2 | 0.26 | 2.5 | 0.31 | 2.9 | 0.74 | 2.9 | 1.03 | 5.5 |
| Brisbane | N | 0 | 0 | 0.03 | 0 | 0.1 | 0 | 0 | 0 | 0.02 | 0 | 0.03 | 0 | 0.01 | 0 | 0.04 | 0 | 0.05 | 0 |
| | S | 0.06 | 0.1 | 0.12 | 0.3 | 0.22 | 1.1 | 0.04 | 0 | 0.06 | 0.1 | 0.07 | 0.2 | 0.06 | 0.3 | 0.13 | 0.4 | 0.17 | 0.4 |
| Melbourne | N | 0 | 0 | 0 | 0 | 0 | 0 | 0 | 0 | 0 | 0 | 0 | 0 | 0 | 0 | 0 | 0 | 0 | 0 |
| | S | 0.28 | 20.6 | 0.02 | 0 | 0.27 | 19.9 | 0 | 0 | 0 | 0 | 0 | 0 | 0 | 0 | 0 | 0 | 0.01 | 0 |

| Protected Case 3 | | Mineral Wool Class 2 | | Class 3 | | Class 4 | | EPS Class 2 | | Class 3 | | Class 4 | | Wood Fibre Class 2 | | Class 3 | | Class 4 | |
|---|---|---|---|---|---|---|---|---|---|---|---|---|---|---|---|---|---|---|---|
| Climate | O | A | B | A | B | A | B | A | B | A | B | A | B | A | B | A | B | A | B |
| Cairns | N | 0.06 | 0.2 | 0.26 | 1.9 | 0.41 | 2.4 | 0.02 | 0 | 0.11 | 1.4 | 0.17 | 1.7 | 0.13 | 1.2 | 0.28 | 2 | 0.42 | 2.3 |
| | S | 0.43 | 3 | 1.24 | 25.3 | 2.81 | 33.8 | 0.33 | 3 | 0.58 | 8.4 | 0.71 | 14.3 | 0.42 | 3.1 | 1.05 | 23.1 | 2.29 | 34 |
| Darwin | N | 0.6 | 2.6 | 1.3 | 6.3 | 1.79 | 14.2 | 0.42 | 2.5 | 0.66 | 2.6 | 0.79 | 2.7 | 0.64 | 2.9 | 1.24 | 4.6 | 1.68 | 13.4 |
| | S | 0.21 | 2.4 | 0.62 | 2.6 | 0.92 | 7.2 | 0.05 | 0.6 | 0.28 | 2.6 | 0.35 | 2.7 | 0.25 | 2.6 | 0.61 | 2.8 | 0.87 | 5.1 |
| Brisbane | N | 0.01 | 0 | 0.03 | 0 | 0.05 | 0 | 0 | 0 | 0.01 | 0 | 0.02 | 0 | 0.03 | 0 | 0.04 | 0 | 0.05 | 0 |
| | S | 0.05 | 0.2 | 0.13 | 0.5 | 0.26 | 0.9 | 0.05 | 0.1 | 0.08 | 0.3 | 0.1 | 0.6 | 0.08 | 0.3 | 0.13 | 0.4 | 0.18 | 0.6 |
| Melbourne | N | 0.01 | 0 | 0 | 0 | 0 | 0 | 0 | 0 | 0 | 0 | 0 | 0 | 0.04 | 0 | 0.02 | 0 | 0.01 | 0 |
| | S | 0.26 | 19.2 | 0.25 | 18.8 | 0.25 | 18.7 | 0 | 0 | 0 | 0 | 0 | 0 | 0.02 | 0 | 0.01 | 0 | 0 | 0 |

The results reveal that corrosion treatment is generally required for all climates, with wood fibre and EPS cases in Melbourne being the only exception. Additionally, it shows that there are only MGI failures for assemblies located in Cairns with low vapour resistant WRBs.

When the CLT assemblies are unprotected with elevated MC during construction, Table 11 for Case 2 shows that the worst-case insulation variation for all climates is EPS. This shows that having vapour resistant insulation material encapsulating the mass timber while the timber has elevated MC is not recommended.

**Table 11.** Unprotected CLT results for split insulation. O (orientation). Performance criteria: A (mould growth index), B (corrosion rate). Classification of WRB: Class 2, 3, and 4 (vapour permeance). Green (pass), red (fail), or yellow (further investigation).

| Unprotected Case 2 Climate | O | Mineral Wool | | | | | | EPS | | | | | | Wood Fibre | | | | | |
|---|---|---|---|---|---|---|---|---|---|---|---|---|---|---|---|---|---|---|---|
| | | Class 2 | | Class 3 | | Class 4 | | Class 2 | | Class 3 | | Class 4 | | Class 2 | | Class 3 | | Class 4 | |
| | | A | B | A | B | A | B | A | B | A | B | A | B | A | B | A | B | A | B |
| Cairns | N | 1.44 | 3.6 | 1.63 | 3.5 | 1.44 | 3.6 | 2.04 | 9.8 | 2.04 | 12 | 2.04 | 9.8 | 1.52 | 3.9 | 1.72 | 4 | 1.52 | 3.9 |
| | S | 2.87 | 23.3 | 2.87 | 37.4 | 3.29 | 40.2 | 3.01 | 34.2 | 2.74 | 30.7 | 2.71 | 28.9 | 2.94 | 29.6 | 2.94 | 41 | 3.52 | 43.8 |
| Darwin | N | 2.33 | 12.6 | 2.46 | 11 | 2.82 | 15.4 | 2.41 | 13.8 | 2.31 | 13.2 | 2.27 | 10.8 | 2.59 | 13 | 2.95 | 12.6 | 3.76 | 16.9 |
| | S | 2.52 | 13.7 | 2.08 | 9.6 | 2.02 | 8.8 | 2.74 | 22.3 | 2.48 | 17.5 | 2.35 | 15.4 | 2.59 | 14.2 | 2.16 | 10.9 | 2.06 | 8.5 |
| Brisbane | N | 1.74 | 10.6 | 1.11 | 2.3 | 0.69 | 1.5 | 2.26 | 22.5 | 2.26 | 19.1 | 2.26 | 18.8 | 1.8 | 11.6 | 1.2 | 2.8 | 0.83 | 1.8 |
| | S | 2.83 | 26.2 | 1.88 | 14.1 | 1.36 | 7.1 | 3.03 | 38.4 | 2.72 | 28.5 | 2.55 | 25.7 | 2.85 | 26.5 | 1.94 | 15.2 | 1.45 | 7.5 |
| Melbourne | N | 1.94 | 15.7 | 1.7 | 10.4 | 0.23 | 0.4 | 2.75 | 31.7 | 2.99 | 30.2 | 2.31 | 26.9 | 2.09 | 16.7 | 0.94 | 2.3 | 0.47 | 1.2 |
| | S | 2.75 | 26.9 | 2.21 | 19 | 0.95 | 24.8 | 3.21 | 59.2 | 2.97 | 41.2 | 2.81 | 37.5 | 2.85 | 27.5 | 1.37 | 8.5 | 0.86 | 7.6 |

For the mineral wool and wood fibre assemblies that are left unprotected during construction, only the temperate and subtropical climates with a low vapour resistant WRB (Class 4) scores a low MGI. This is because the CLT has a greater drying potential once enclosed if WRBs with high vapour resistance are avoided. Please refer to Appendix B for full unprotected case results.

### 5.3. Internal Insulation Cases

The third analysis concerns the internal insulation solutions, as depicted in Figure 8. The performance criteria result for protected assemblies are presented in Table 12. Full results can be found in Appendices A and B.

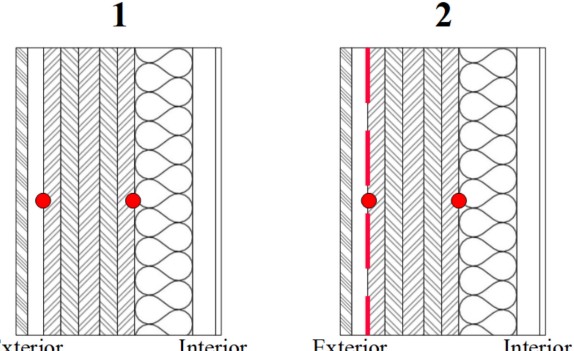

**Figure 8.** Wall assembly with internal insulation. No WRB modelled (Case 1), and WRB adhered to exterior of CLT panel (Case 2). Red points represent the monitoring positions at the internal and external surface of the CLT panel.

The results in Table 12 show that, for protected assemblies with internal insulation, only the mineral wool and wood fibre assemblies located in Melbourne fail the MGI threshold. While assemblies in Cairns also fail if no membrane is installed. The corrosion rate criterion is satisfied for both orientations only in the subtropical climate when a high vapour resistant WRB and vapour resistant insulation material is installed. For all other cases, coatings or alloys for corrosion treatment should be considered for metal elements.

In tropical climates where the insulation is located internally, the cooled internal air mass creates a temperature gradient across the assembly where the dew point is less likely to occur at the surface of the CLT panel, when compared to assemblies with external insulation. This is because when the CLT panel is on the external side of the insulation layer, it is kept at a higher temperature due to the ambient conditions, consequently decreasing the relative humidity within the CLT layer materials. This has the effect of decreasing the MGI for internal insulation solutions.

If internal insulation assemblies are unprotected during construction (results shown in Table 13) both moisture-safety criteria are exceeded for the temperate climate. It should also be noted that the MGI is high for the subtropical and tropical climates though is not exceeded.

**Table 12.** Protected CLT results for internal insulation. O (orientation). Performance criteria: A (mould growth index), B (corrosion rate). Classification of WRB: Class 2, 3, and 4 (vapour permeance). Green (pass), red (fail), or yellow (further investigation).

| Protected Case 1 | | Mineral Wool No WRB | | | EPS No WRB | | | Wood Fibre No WRB | |
|---|---|---|---|---|---|---|---|---|---|
| Climate | O | A | B | | A | B | | A | B |
| Cairns | N | 0.37 | 1.3 | | 0.37 | 1.4 | | 0.37 | 1.4 |
| | S | 4.11 | 33.1 | | 4.13 | 34.3 | | 4.08 | 33 |
| Darwin | N | 1.84 | 8.4 | | 1.82 | 8.6 | | 1.85 | 8.4 |
| | S | 0.62 | 1.9 | | 0.61 | 1.8 | | 0.62 | 1.9 |
| Brisbane | N | 0.2 | 0 | | 0.2 | 0 | | 0.2 | 0 |
| | S | 0.5 | 6.7 | | 0.51 | 3.8 | | 0.49 | 3.6 |
| Melbourne | N | 0.12 | 3.5 | | 0.02 | 0 | | 0.02 | 0 |
| | S | 3.43 | 61.5 | | 0.16 | 3 | | 2.61 | 54.2 |

| Protected Case 2 | | Mineral Wool Class 2 | | Class 3 | | Class 4 | | EPS Class 2 | | Class 3 | | Class 4 | | Wood Fibre Class 2 | | Class 3 | | Class 4 | |
|---|---|---|---|---|---|---|---|---|---|---|---|---|---|---|---|---|---|---|---|
| Climate | O | A | B | A | B | A | B | A | B | A | B | A | B | A | B | A | B | A | B |
| Cairns | N | 0 | 0 | 0.06 | 0 | 0.18 | 0.4 | 0 | 0 | 0.06 | 0 | 0.18 | 0.3 | 0 | 0 | 0.06 | 0 | 0.18 | 0.4 |
| | S | 0.24 | 2.3 | 0.9 | 17.7 | 2.71 | 29.5 | 0.25 | 2.4 | 1.01 | 19.1 | 2.76 | 30.8 | 0.23 | 2.3 | 0.9 | 17.7 | 2.7 | 29.5 |
| Darwin | N | 0.35 | 2.2 | 0.94 | 2.8 | 1.43 | 6.3 | 0.37 | 2.3 | 0.94 | 3 | 1.41 | 6.5 | 0.34 | 2.2 | 0.94 | 2.8 | 1.44 | 6.4 |
| | S | 0.01 | 0 | 0.12 | 0.6 | 0.37 | 1.4 | 0.01 | 0 | 0.12 | 0.7 | 0.35 | 1.4 | 0.01 | 0 | 0.13 | 0.7 | 0.37 | 1.5 |
| Brisbane | N | 0 | 0 | 0.01 | 0 | 0.07 | 0 | 0 | 0 | 0.01 | 0 | 0.07 | 0 | 0.01 | 0 | 0.01 | 0 | 0.07 | 0 |
| | S | 0.29 | 6.6 | 0.3 | 6.5 | 0.3 | 6.5 | 0.03 | 0 | 0.09 | 0.1 | 0.19 | 2.7 | 0.11 | 1.3 | 0.11 | 1.2 | 0.17 | 2.3 |
| Melbourne | N | 0.12 | 3.8 | 0.12 | 3.5 | 0.11 | 3.4 | 0 | 0 | 0 | 0 | 0 | 0 | 0 | 0 | 0 | 0 | 0 | 0 |
| | S | 3.45 | 62.2 | 3.43 | 61.5 | 3.42 | 61.4 | 0.13 | 3.5 | 0.11 | 3 | 0.1 | 2.9 | 2.66 | 55.2 | 2.61 | 54.3 | 2.6 | 54.1 |

**Table 13.** Unprotected CLT results for internal insulation. O (orientation). Performance criteria: A (mould growth index), B (corrosion rate). Classification of WRB: Class 2, 3, and 4 (vapour permeance). Green (pass), red (fail), or yellow (further investigation).

| Unprotected Case 2 | | Mineral Wool Class 2 | | Class 3 | | Class 4 | | EPS Class 2 | | Class 3 | | Class 4 | | Wood Fibre Class 2 | | Class 3 | | Class 4 | |
|---|---|---|---|---|---|---|---|---|---|---|---|---|---|---|---|---|---|---|---|
| Climate | O | A | B | A | B | A | B | A | B | A | B | A | B | A | B | A | B | A | B |
| Cairns | N | 1.74 | 5.6 | 1.32 | 2.5 | 1.07 | 2 | 2.71 | 18.8 | 2.71 | 16.5 | 2.71 | 15.4 | 1.73 | 5.9 | 1.32 | 2.5 | 1.07 | 2 |
| | S | 2.71 | 17.7 | 2.36 | 25.9 | 2.86 | 33.1 | 2.73 | 34.3 | 2.62 | 31.1 | 3.01 | 35.9 | 2.71 | 18.2 | 2.39 | 25.9 | 2.87 | 33.1 |
| Darwin | N | 2.01 | 3.6 | 2 | 3.8 | 2.17 | 7.1 | 2.71 | 17.3 | 2.71 | 15.7 | 2.7 | 15.2 | 2.01 | 5 | 2 | 3.9 | 2.17 | 7.2 |
| | S | 2.17 | 12.1 | 1.54 | 3.4 | 1.3 | 2.7 | 2.59 | 20.1 | 2.59 | 16.9 | 2.58 | 16.3 | 2.16 | 12.3 | 1.54 | 3.4 | 1.3 | 2.7 |
| Brisbane | N | 1.47 | 6.4 | 0.88 | 1.8 | 0.42 | 0.8 | 2.78 | 25.1 | 2.78 | 19.6 | 2.78 | 19.1 | 1.47 | 6.5 | 0.88 | 1.8 | 0.42 | 0.8 |
| | S | 2.72 | 26.6 | 1.76 | 13.3 | 1.16 | 9.1 | 2.74 | 40.5 | 2.73 | 29.7 | 2.73 | 28.5 | 2.73 | 26.8 | 1.77 | 13.5 | 1.16 | 8.1 |
| Melbourne | N | 0.1 | 2.1 | 0.11 | 2.3 | 0.11 | 2.3 | 2.95 | 39.8 | 2.95 | 29.6 | 2.95 | 28.8 | 1.07 | 9 | 0.99 | 7.7 | 1 | 7.2 |
| | S | 3.56 | 57.2 | 3.6 | 56.3 | 3.61 | 55.9 | 2.94 | 81.2 | 2.94 | 48.5 | 2.94 | 47.2 | 3.07 | 66.1 | 3.07 | 63.1 | 3.07 | 62.3 |

## 6. Recommendations

Recommendations for CLT assembly solutions in hot and humid Australian climates are provided in this section. These are provided to assist building designers construct energy-efficient and moisture-safe mass timber buildings for Australian climates, while advancing Australian building practices during the design and construction phases. Assembly solutions are groups by climate, as results for the tropical climates of Cairns and Darwin showed similarity due to inward vapour drive for most of the year, while the subtropical and temperate climates, Brisbane, and Melbourne, showed similarity due to their predominant outwards vapour drive.

### 6.1. Tropical Climate Zones Construction Solutions

Recommendation for mass timber construction solutions in tropical climates, such as Cairns and Darwin can be categorised by the location of the insulation, as below:

- *Exterior insulation* can be used in these climates; however, it is recommended that either vapour impermeable insulation or a WRB with high vapour resistance is installed externally to avoid mould growth on the external surface of the CLT panel. It is critical that they are only installed over dry CLT panels, while the WRB also functions as an airtightness control layer (refer to Tables 8 and 9).
- *Internal insulation* can perform well in hot and humid climates. The use of a WRB with high vapour resistance on the external side of the CLT panel is required to reduce the amount of moisture moving into the assembly. Again, it is highly recommended to

install the impermeable membrane only over dry CLT panels, and that the WRB also functions as an airtightness control layer (refer to Tables 10 and 11).

- *Split insulation* solutions should follow the same recommendations provided for the exterior insulation (refer to Tables 12 and 13).

General recommendations for CLT assemblies in hot and humid climates include:

- Vapour impermeable insulation or internal finishes installed internally should be avoided. This is because moisture that ingresses into the assembly, may accumulate without a path to dry out in the internal direction.
- Before installing assembly materials externally with high vapour resistance, it should be ensured that the CLT panels are dry.
- The use of drained and ventilated non-absorptive rainscreen cladding is highly recommended.

### 6.2. Subtropical and Temperate Climate Zones Construction Solutions

Recommendation for mass timber construction solutions in subtropical and temperate climates, such as Brisbane and Melbourne can be categorised by the location of the insulation, as below:

- *External insulation* is the suggested solution in subtropical and temperate climate zones. Vapour permeable insulation and adhered membranes are recommended unless dry CLT panels can be ensured (refer to Tables 8 and 9).
- *Internal insulation* is generally not recommended particularly in temperate and cold climates, because controlling thermal bridging with internal insulation is complex and may not always be possible. Though internal insulation may be possible in subtropical climates with high vapour resistant WRBs (refer to Tables 10 and 11).
- *Split insulation* guidance is the same as for the exterior insulation (refer to Tables 12 and 13).

General recommendations for CLT assemblies in subtropical and temperate climates include:

- Vapour impermeable insulation or WRBs with high vapour resistance installed externally should be avoided. This is because moisture that ingresses into the assembly, may accumulate without a path to dry out in the external direction.
- The use of drained and ventilated non-absorptive rainscreen cladding is highly recommended.
- These strategies may allow for durable CLT wall construction without the need for surface treatment of the panels when combined with both good moisture management and PH standard levels of airtightness, and mechanically controlled ventilation (with energy recovery and dehumidification where appropriate). In addition, they align with principals for low-energy buildings, resulting in high occupant comfort and indoor environmental quality.

### 6.3. Design-Phase Recommendations

Simulation results suggest that, when employing either exterior, interior, or split insulation solutions, corrosion of fixings cannot be avoided for buildings located in tropical and subtropical climates. Therefore, measures should be taken to protect metal connectors or cladding fixtures from corrosion risk if not clean iron. Such measures may include corrosion protection topcoats or a moisture resistant alloy.

When considering the suitability of the external insulation, insulation materials with water shedding properties on the external surface should be employed, additionally the insulation boards should be installed flush to prevent wind-washing when a WRB is not installed outboard of the insulation. The WRB should be taped and sealed with cladding fixtures flashed to prevent moisture from wicking past the insulation layer. This is because, while cladding should be designed to act as the primary rain barrier, laboratory studies found there is a possibility of driving rain finding a path through edge profiles and cladding fixings [70]. The same study found that approximately 0.5% of the driving rain passed the

cladding and may run down the external face of the insulation layer, coming in contact with the cladding fixings under the force of gravity or air currents. Capillary suction may then force rain leakage through the bolt or screw hole, providing the panels with a source of moisture for a prolonged period. It should be noted that a well-constructed, airtight WRB and a ventilated cavity, by way of top and bottom vents located at designed intervals, will create a pressure equalised system. This reduces the relative negative air pressure on the interior of the building, preventing bulk water being pulled into the assembly [71]. A ventilated cavity and WRB, therefore, adds moisture-safety redundancy for the assembly design.

Façade and mechanical engineers must work collaboratively to ensure the design intent is achieved for a durable and comfortable design where interior relative humidity is maintained below 70%. This may require the use of air conditioning system with integrated dehumidification function, such as highly efficient split air conditioners that use waste-heat to reduce latent load with a re-heat function, wrap-around heat pipes, or high-quality stand-alone dehumidification units.

*6.4. Construction-Phase Recommendations*

It is recommended to plan for moisture management early, with well-defined procedures for good storm-water management to protect assemblies from wetting during construction. This may include performing a risk analysis of any element of the CLT panels exposure to moisture, along with construction strategies to provide an airtight WRB: taping over the seams of the WRB with solid acrylic tape, sealing all CLT panel end-grains before reaching the construction site, removing any pooling of water, and considering tenting construction methods. Moisture content of CLT panels should be intermittently checked or continuously monitored with sensors, especially if exposed to pooling rain or rain events. Installing a fan heater may also be considered to reduce the CLT panel moisture content to less than 80% RH before cladding internally. Alternatively, prefabrication of the assemblies can reduce the risks of moisture ingress occurring during construction.

The results of the study presented in this paper show that CLT assemblies with WRBs offer a more resilient solution for moisture risk management. However, this assumes that the WRB can maintain a barrier against driving rain during construction and operation. This is more likely the case when the WRB is adhered to the CLT surface during or prior to construction. As a watertight and airtight seal can be more easily ensured when adhering a WRB to a rigid surface, such as the CLT panel, compared to a frame batten construction. This is especially the case for assemblies with high wind loads, such as those placed on multi-storey buildings. Adhering to a rigid surface offers the maximum level of security regarding airtightness and ensures that moisture transfer due to convective effects into the panel is eliminated. It is recommended that any seams in the WRB should overlap from above, to ensure moisture drains away from the CLT panel.

There are also several issues that should be considered for WRB in the Australian climates; for example, tropical monsoon conditions may cause de-adhesion, while UV damage may cause deterioration of the WRB. For these reasons, the construction-phase should be as short as possible, so that the WRB can be protected by the thermal layer, avoiding variations of temperature, exposure to UV, and direct driving rain.

## 7. Limitations

The literature shows that WUFI demonstrates agreement with empirical tests, even though several limitations exist. WUFI does not consider sorption hysteresis where prior conditions of moisture exposure impact future conditions [72]; an estimation is used to generate the liquid transport coefficients [73], and the hygrothermal simulations assume that the materials have constant geometry with no swelling or shrinkage, no chemical reactions, no change in the material properties by damage or aging, and the moisture storage function is treated as independent of temperature [74]. Håkansson [75] describes a sorption effect for a Fickian model, as WUFI employs, which retards moisture at high RHs,

leading to an over-estimation of moisture penetrating through wood when modelled with a dynamic sorption curve.

The WUFI Pro software version has been utilised in this study, which performs one-dimensional transient simulation on building component cross-sections and is not able to account for construction discontinuities in two or three dimensions. Construction discontinuities have a greater risk of moisture penetration (in liquid or vapour form) at these locations, such as at window and balcony junctions.

In this study, only the internal and external surfaces of the CLT panel were tested against the ASHRAE 160 moisture safety criteria. It is possible that moisture issues may emerge at other locations in the assembly, however, these locations were out of the scope of this paper.

## 8. Conclusions

This paper investigated design and construction features of highly energy-efficient CLT buildings in Australian subtropical and tropical climates; results may inform climate-specific design to minimise mould risks, specifically in respect to the position of the insulation layer, vapour permeability of the WRB, and influence of good stormwater management.

The main findings of this study, which employed a step-by-step parametric analysis within WUFI, indicate that CLT structures built to the PH standard can provide excellent performance in Australian hot and humid climates, but climate specific considerations need to be made to ensure proper design, construction, and maintenance to avoid conditions leading to degradation.

The results show that sufficient drying capacity can be maintained and wetting minimised to avoid moisture risks by identifying moisture-safe assemblies and interior ventilation practices specific to the four selected Australian climates. The hygrothermal study evaluated the performance of external, internal, and split insulation solutions, with three off-the-shelf WRBs. The north and south orientation were simulated with 1% of driving rain deposited on the material directly facing the ventilated air cavity to represent any imperfections in the cladding during operation.

For mass timber buildings located in tropical environments, where the levels of ambient vapour pressure are significant and a predominantly cooled interior environment is expected, results of the hygrothermal parametric analysis suggest that common moisture-open envelope design strategies with highly permeable membranes may not be the optimal solution. The location of the insulation layer (externally, internally or split between external and internal) was shown to not exceed the mould growth criteria for tropical climates where a WRB of Class 2 vapour permeance is adhered to the CLT panel. However, this solution can only be taken in conjunction with adequate procedures for storm-water management. Additionally, it was concluded that the use of high vapour resistant insulation, such as EPS, in the CLT walls should be undertaken with caution and only with effective supervision for installation of over dry CLT panels.

**Author Contributions:** Conceptualization and methodology, P.L., M.S., A.B., E.G.; formal analysis, M.S.; data curation, M.S.; writing, M.S.; reviewing, P.L., A.B. and E.G.; visualization, M.S.; supervision, P.L.; funding acquisition, P.L. All authors have read and agreed to the published version of the manuscript.

**Funding:** This research was supported by the Australian Research Council Research Hub for Advanced Solutions to Transform Tall Timber Buildings (project number IH150100030), funded by the Australian Government.

**Institutional Review Board Statement:** Not applicable.

**Informed Consent Statement:** Not applicable.

**Data Availability Statement:** Data sharing not applicable to this article.

**Conflicts of Interest:** The authors declare that there is no conflict of interest.

## Appendix A. Summary of Protected CLT Results

**Table A1.** Protected CLT results for external insulation. O (orientation). Performance criteria: A (mould growth index), B (corrosion rate). Classification of WRB: Class 2, 3, and 4 (vapour permeance). Green (pass), red (fail), or yellow (further investigation). Critical surface that fails the performance criteria indicated in right table: Ext. (external surface), Int. (internal surface), Both (both surfaces).

### Protected Case 1

| | | Performance Criteria | | | | | | Critical Surface | | |
|---|---|---|---|---|---|---|---|---|---|---|
| | | Mineral Wool No WRB | | EPS No WRB | | Wood Fibre No WRB | | Mineral Wool No WRB | EPS No WRB | Wood Fibre No WRB |
| Climate | O | A | B | A | B | A | B | | | |
| Cairns | N | 0.78 | 3.9 | 0.15 | 1.9 | 0.65 | 3.8 | Ext. | Ext. | Ext. |
| Cairns | S | 3.76 | 42.9 | 0.47 | 3.19 | 3.56 | 43.6 | Ext. | Ext. | Ext. |
| Darwin | N | 2.73 | 29.9 | 0.55 | 2.7 | 3.24 | 24.6 | Ext. | Ext. | Ext. |
| Darwin | S | 1.45 | 22.5 | 0.3 | 2.7 | 1.24 | 12.7 | Ext. | Ext. | Ext. |
| Brisbane | N | 0.14 | 0 | 0.03 | 0 | 0.07 | 0 | Ext. | - | - |
| Brisbane | S | 0.31 | 1.5 | 0.07 | 0.3 | 0.2 | 0.6 | Ext. | Ext. | Ext. |
| Melbourne | N | 0 | 0 | 0 | 0 | 0.01 | 0 | Ext. | - | - |
| Melbourne | S | 0.06 | 0 | 0 | 0 | 0 | 0 | Ext. | - | - |

### Protected Case 2

| | | Mineral Wool | | | | | | EPS | | | | | | Wood Fibre | | | | | | Mineral Wool | | | EPS | | | Wood Fibre | | |
|---|---|---|---|---|---|---|---|---|---|---|---|---|---|---|---|---|---|---|---|---|---|---|---|---|---|---|---|---|
| | | Class 2 | | Class 3 | | Class 4 | | Class 2 | | Class 3 | | Class 4 | | Class 2 | | Class 3 | | Class 4 | | Cla. 2 | Cla. 3 | Cla. 4 | Cla. 2 | Cla. 3 | Cla. 4 | Cla. 2 | Cla. 3 | Cla. 4 |
| Climate | O | A | B | A | B | A | B | A | B | A | B | A | B | A | B | A | B | A | B | | | | | | | | | |
| Cairns | N | 0.06 | 1.1 | 0.48 | 2.7 | 0.68 | 3.8 | 0.15 | 1.9 | 0.12 | 1.8 | 0.15 | 1.9 | 0.61 | 3.6 | 0.47 | 2.9 | 0.61 | 3.6 | Ext. | Ext. | Ext. | Ext. | Ext. | Ext. | Ext. | Ext. | Ext. |
| Cairns | S | 0.66 | 8.9 | 2.2 | 34.4 | 3.39 | 41.5 | 0.27 | 2.9 | 0.41 | 3.1 | 0.46 | 3.1 | 0.71 | 14.4 | 2.25 | 35.5 | 3.27 | 42.5 | Ext. | Ext. | Ext. | Ext. | Ext. | Ext. | Ext. | Ext. | Ext. |
| Darwin | N | 0.75 | 3 | 1.54 | 14.6 | 2.16 | 28.3 | 0.35 | 2.6 | 0.49 | 2.7 | 0.54 | 2.7 | 1 | 4.1 | 1.99 | 11.6 | 2.82 | 20.3 | Ext. | Ext. | Ext. | Ext. | Ext. | Ext. | Ext. | Ext. | Ext. |
| Darwin | S | 0.46 | 2.9 | 1.02 | 9.2 | 1.34 | 18.2 | 0.1 | 1.6 | 0.25 | 2.6 | 0.29 | 2.7 | 0.46 | 3.3 | 0.91 | 3.4 | 1.16 | 11.1 | Ext. | Ext. | Ext. | Ext. | Ext. | Ext. | Ext. | Ext. | Ext. |
| Brisbane | N | 0.03 | 0 | 0.05 | 0 | 0.08 | 0 | 0.01 | 0 | 0.03 | 0 | 0.03 | 0 | 0.03 | 0 | 0.06 | 0.1 | 0.07 | 0.1 | Ext. | Ext. | Ext. | - | - | - | - | Ext. | Ext. |
| Brisbane | S | 0.08 | 0.4 | 0.14 | 0.4 | 0.23 | 1 | 0.04 | 0.1 | 0.06 | 0.3 | 0.07 | 0.3 | 0.09 | 0.6 | 0.16 | 0.7 | 0.19 | 0.6 | Ext. | Ext. | Ext. | Ext. | Ext. | Ext. | Ext. | Ext. | Ext. |
| Melbourne | N | 0 | 0 | 0 | 0 | 0 | 0 | 0 | 0 | 0 | 0 | 0 | 0 | 0 | 0 | 0.01 | 0 | 0.01 | 0 | Ext. | Ext. | Ext. | - | - | - | - | - | - |
| Melbourne | S | 0 | 0 | 0.01 | 0 | 0.04 | 0 | 0 | 0 | 0 | 0 | 0 | 0 | 0 | 0 | 0 | 0 | 0 | 0 | Ext. | Ext. | Ext. | - | - | - | - | - | - |

### Protected Case 3

| | | Performance Criteria | | | | | | | | | | | | | | | | | | Critical Surface | | | | | | | | |
|---|---|---|---|---|---|---|---|---|---|---|---|---|---|---|---|---|---|---|---|---|---|---|---|---|---|---|---|---|
| | | Mineral Wool | | | | | | EPS | | | | | | Wood Fibre | | | | | | Mineral Wool | | | EPS | | | Wood Fibre | | |
| | | Class 2 | | Class 3 | | Class 4 | | Class 2 | | Class 3 | | Class 4 | | Class 2 | | Class 3 | | Class 4 | | Cla. 2 | Cla. 3 | Cla. 4 | Cla. 2 | Cla. 3 | Cla. 4 | Cla. 2 | Cla. 3 | Cla. 4 |
| Climate | O | A | B | A | B | A | B | A | B | A | B | A | B | A | B | A | B | A | B | | | | | | | | | |
| Cairns | N | 0.18 | 2 | 0.47 | 2.2 | 0.65 | 3.6 | 0.05 | 0.9 | 0.17 | 1.9 | 0.2 | 2 | 0.31 | 2.8 | 0.47 | 2.5 | 0.59 | 3.2 | Ext. | Ext. | Ext. | Ext. | Ext. | Ext. | Ext. | Ext. | Ext. |
| Cairns | S | 0.56 | 4.9 | 1.52 | 32.4 | 2.95 | 40 | 0.34 | 3.1 | 0.51 | 3.6 | 0.58 | 5.1 | 0.53 | 3.4 | 1.04 | 26.3 | 2.1 | 37.3 | Ext. | Ext. | Ext. | Ext. | Ext. | Ext. | Ext. | Ext. | Ext. |
| Darwin | N | 0.73 | 2.8 | 1.48 | 14.9 | 2.01 | 28.9 | 0.4 | 2.6 | 0.56 | 2.8 | 0.62 | 2.8 | 0.82 | 3.6 | 1.33 | 8.3 | 1.72 | 19.4 | Ext. | Ext. | Ext. | Ext. | Ext. | Ext. | Ext. | Ext. | Ext. |
| Darwin | S | 0.4 | 2.8 | 0.93 | 10.1 | 1.22 | 20.8 | 0.17 | 2.4 | 0.3 | 2.7 | 0.35 | 2.8 | 0.43 | 2.8 | 0.79 | 3 | 1.04 | 10.9 | Ext. | Ext. | Ext. | Ext. | Ext. | Ext. | Ext. | | |
| Brisbane | N | 0.03 | 0 | 0.04 | 0 | 0.06 | 0 | 0.01 | 0 | 0.03 | 0 | 0.03 | 0 | 0.07 | 0.3 | 0.07 | 0.2 | 0.07 | 0.1 | Ext. | Ext. | Ext. | - | - | - | - | Ext. | Ext. |
| Brisbane | S | 0.12 | 0.4 | 0.2 | 0.7 | 0.31 | 1.3 | 0.06 | 0.3 | 0.08 | 0.4 | 0.08 | 0.4 | 0.13 | 0.8 | 0.17 | 0.8 | 0.21 | 0.7 | Ext. | Ext. | Ext. | Ext. | Ext. | Ext. | Ext. | Ext. | Ext. |
| Melbourne | N | 0.01 | 0 | 0 | 0 | 0 | 0 | 0 | 0 | 0 | 0 | 0 | 0 | 0.06 | 0.3 | 0.02 | 0 | 0.02 | 0 | Ext. | Ext. | Ext. | - | - | - | Ext. | - | - |
| Melbourne | S | 0 | 0 | 0 | 0 | 0.01 | 0 | 0 | 0 | 0 | 0 | 0 | 0 | 0.02 | 0 | 0.01 | 0 | 0 | 0 | Ext. | Ext. | Ext. | - | - | - | - | - | - |

**Table A2.** Protected CLT results for split insulation. O (orientation). Performance criteria: A (mould growth index), B (corrosion rate). Classification of WRB: Class 2, 3, and 4 (vapour permeance). Green (pass), red (fail), or yellow (further investigation). Critical surface that fails the performance criteria indicated in right table: Ext. (external surface), Int. (internal surface), Both (both surfaces).

**Protected Case 1**

| Climate | O | Mineral Wool No WRB A | B | EPS No WRB A | B | Wood Fibre No WRB A | B | Mineral Wool No WRB | EPS No WRB | Wood Fibre No WRB |
|---|---|---|---|---|---|---|---|---|---|---|
| Cairns | N | 0.53 | 3 | 0.1 | 0.7 | 0.49 | 3.2 | Ext. | Ext. | Ext. |
| | S | 3.86 | 35.6 | 0.55 | 6.9 | 3.98 | 38.3 | Ext. | Ext. | Ext. |
| Darwin | N | 2.15 | 15.2 | 0.68 | 2.6 | 3.71 | 15.4 | Ext. | Ext. | Ext. |
| | S | 1.15 | 8.1 | 0.27 | 2.5 | 1.11 | 7.2 | Ext. | Ext. | Ext. |
| Brisbane | N | 0.18 | 0 | 0.03 | 0 | 0.08 | 0 | Ext. | - | - |
| | S | 0.4 | 1.7 | 0.07 | 0.2 | 0.19 | 0.5 | Ext. | Ext. | Ext. |
| Melbourne | N | 0 | 0 | 0 | 0 | 0 | 0 | Ext. | - | - |
| | S | 0.27 | 20.2 | 0 | 0 | 0.01 | 0 | Int. | - | - |

**Protected Case 2**

| Climate | O | Mineral Wool Class 2 A | B | Class 3 A | B | Class 4 A | B | EPS Class 2 A | B | Class 3 A | B | Class 4 A | B | Wood Fibre Class 2 A | B | Class 3 A | B | Class 4 A | B | MW Cla.2 | Cla.3 | Cla.4 | EPS Cla.2 | Cla.3 | Cla.4 | WF Cla.2 | Cla.3 | Cla.4 |
|---|---|---|---|---|---|---|---|---|---|---|---|---|---|---|---|---|---|---|---|---|---|---|---|---|---|---|---|---|
| Cairns | N | 0.43 | 2.8 | 0.28 | 1.8 | 0.43 | 2.8 | 0.09 | 0.6 | 0.05 | 0 | 0.09 | 0.6 | 0.44 | 3 | 0.32 | 2 | 0.44 | 3 | Ext. | Ext. | Ext. | Ext. | - | Ext. | Ext. | Ext. | Ext. |
| | S | 0.5 | 3.9 | 1.65 | 27.5 | 3.28 | 34.6 | 0.22 | 2.4 | 0.44 | 3.1 | 0.53 | 5.8 | 0.61 | 10.4 | 2.14 | 31.4 | 3.46 | 37.3 | Ext. | Ext. | Ext. | Ext. | Ext. | Ext. | Ext. | Ext. | Ext. |
| Darwin | N | 0.64 | 2.7 | 1.39 | 7.1 | 1.91 | 13.8 | 0.35 | 2.4 | 0.57 | 2.6 | 0.66 | 2.6 | 0.91 | 3.5 | 1.87 | 8.2 | 2.93 | 14.2 | Ext. | Ext. | Ext. | Ext. | Ext. | Ext. | Ext. | Ext. | Ext. |
| | S | 0.27 | 2.1 | 0.71 | 2.6 | 1.05 | 6.6 | 0.02 | 0 | 0.2 | 2.2 | 0.26 | 2.5 | 0.31 | 2.9 | 0.74 | 2.9 | 1.03 | 5.5 | Ext. | Ext. | Ext. | - | Ext. | Ext. | Ext. | Ext. | Ext. |
| Brisbane | N | 0 | 0 | 0.03 | 0 | 0.1 | 0 | 0 | 0 | 0.02 | 0 | 0.03 | 0 | 0.01 | 0 | 0.04 | 0 | 0.05 | 0 | Ext. | Ext. | Ext. | - | - | - | - | - | - |
| | S | 0.06 | 0.1 | 0.12 | 0.3 | 0.22 | 1.1 | 0.04 | 0 | 0.06 | 0.1 | 0.07 | 0.2 | 0.06 | 0.3 | 0.13 | 0.4 | 0.17 | 0.4 | Ext. | Ext. | Ext. | - | Ext. | Ext. | Ext. | Ext. | Ext. |
| Melbourne | N | 0 | 0 | 0 | 0 | 0 | 0 | 0 | 0 | 0 | 0 | 0 | 0 | 0 | 0 | 0 | 0 | 0 | 0 | Ext. | Ext. | Ext. | - | - | - | - | - | - |
| | S | 0.28 | 20.6 | 0.02 | 0 | 0.27 | 19.9 | 0 | 0 | 0 | 0 | 0 | 0 | 0 | 0 | 0 | 0 | 0.01 | 0 | Int. | Ext. | Int. | - | - | - | - | - | - |

**Protected Case 3**

| Climate | O | Mineral Wool Class 2 A | B | Class 3 A | B | Class 4 A | B | EPS Class 2 A | B | Class 3 A | B | Class 4 A | B | Wood Fibre Class 2 A | B | Class 3 A | B | Class 4 A | B | MW Cla.2 | Cla.3 | Cla.4 | EPS Cla.2 | Cla.3 | Cla.4 | WF Cla.2 | Cla.3 | Cla.4 |
|---|---|---|---|---|---|---|---|---|---|---|---|---|---|---|---|---|---|---|---|---|---|---|---|---|---|---|---|---|
| Cairns | N | 0.06 | 0.2 | 0.26 | 1.9 | 0.41 | 2.4 | 0.02 | 0 | 0.11 | 1.4 | 0.17 | 1.7 | 0.13 | 1.2 | 0.28 | 2 | 0.42 | 2.3 | Ext. | Ext. | Ext. | - | Ext. | Ext. | Ext. | Ext. | Ext. |
| | S | 0.43 | 3 | 1.24 | 25.3 | 2.81 | 33.8 | 0.33 | 3 | 0.58 | 8.4 | 0.71 | 14.3 | 0.42 | 3.1 | 1.05 | 23.1 | 2.29 | 34 | Ext. | Ext. | Ext. | Ext. | Ext. | Ext. | Ext. | Ext. | Ext. |
| Darwin | N | 0.6 | 2.6 | 1.3 | 6.3 | 1.79 | 14.2 | 0.42 | 2.5 | 0.66 | 2.6 | 0.79 | 2.7 | 0.64 | 2.9 | 1.24 | 4.6 | 1.68 | 13.4 | Ext. | Ext. | Ext. | Ext. | Ext. | Ext. | Ext. | Ext. | Ext. |
| | S | 0.21 | 2.4 | 0.62 | 2.6 | 0.92 | 7.2 | 0.05 | 0.6 | 0.28 | 2.6 | 0.35 | 2.7 | 0.25 | 2.6 | 0.61 | 2.8 | 0.87 | 5.1 | Ext. | Ext. | Ext. | Ext. | Ext. | Ext. | Ext. | Ext. | Ext. |
| Brisbane | N | 0.01 | 0 | 0.03 | 0 | 0.05 | 0 | 0 | 0 | 0.01 | 0 | 0.02 | 0 | 0.03 | 0 | 0.04 | 0 | 0.05 | 0 | Ext. | Ext. | Ext. | - | - | - | - | - | - |
| | S | 0.05 | 0.2 | 0.13 | 0.5 | 0.26 | 0.9 | 0.05 | 0.1 | 0.08 | 0.3 | 0.1 | 0.6 | 0.08 | 0.3 | 0.13 | 0.4 | 0.18 | 0.6 | Ext. | Ext. | Ext. | Ext. | Ext. | Ext. | Ext. | Ext. | Ext. |
| Melbourne | N | 0.01 | 0 | 0 | 0 | 0 | 0 | 0 | 0 | 0 | 0 | 0 | 0 | 0.04 | 0 | 0.02 | 0 | 0.01 | 0 | Ext. | Ext. | Ext. | - | - | - | - | - | - |
| | S | 0.26 | 19.2 | 0.25 | 18.8 | 0.25 | 18.7 | 0 | 0 | 0 | 0 | 0 | 0 | 0.02 | 0 | 0.01 | 0 | 0 | 0 | Int. | Int. | Int. | - | - | - | - | - | - |

**Table A3.** Protected CLT results for internal insulation. O (orientation). Performance criteria: A (mould growth index), B (corrosion rate). Classification of WRB: Class 2, 3, and 4 (vapour permeance). Green (pass), red (fail), or yellow (further investigation). Critical surface that fails the performance criteria indicated in right table: Ext. (external surface), Int. (internal surface), Both (both surfaces).

| Protected Case 1 | | Mineral Wool No WRB | | EPS No WRB | | Wood Fibre No WRB | | Mineral Wool No WRB | EPS No WRB | Wood Fibre No WRB |
|---|---|---|---|---|---|---|---|---|---|---|
| Climate | O | A | B | A | B | A | B | | | |
| Cairns | N | 0.37 | 1.3 | 0.37 | 1.4 | 0.37 | 1.4 | Ext. | Ext. | Ext. |
| | S | 4.11 | 33.1 | 4.13 | 34.3 | 4.08 | 33 | Ext. | Ext. | Ext. |
| Darwin | N | 1.84 | 8.4 | 1.82 | 8.6 | 1.85 | 8.4 | Ext. | Ext. | Ext. |
| | S | 0.62 | 1.9 | 0.61 | 1.8 | 0.62 | 1.9 | Ext. | Ext. | Ext. |
| Brisbane | N | 0.2 | 0 | 0.2 | 0 | 0.2 | 0 | Ext. | - | - |
| | S | 0.5 | 6.7 | 0.51 | 3.8 | 0.49 | 3.6 | Both | Ext. | Ext. |
| Melbourne | N | 0.12 | 3.5 | 0.02 | 0 | 0.02 | 0 | Int. | - | - |
| | S | 3.43 | 61.5 | 0.16 | 3 | 2.61 | 54.2 | Int. | Both | Int. |

| Protected Case 2 | | Mineral Wool Class 2 | | Class 3 | | Class 4 | | EPS Class 2 | | Class 3 | | Class 4 | | Wood Fibre Class 2 | | Class 3 | | Class 4 | | MW Cla. 2 | Cla. 3 | Cla. 4 | EPS Cla. 2 | Cla. 3 | Cla. 4 | WF Cla. 2 | Cla. 3 | Cla. 4 |
|---|---|---|---|---|---|---|---|---|---|---|---|---|---|---|---|---|---|---|---|---|---|---|---|---|---|---|---|---|
| Climate | O | A | B | A | B | A | B | A | B | A | B | A | B | A | B | A | B | A | B | | | | | | | | | |
| Cairns | N | 0 | 0 | 0.06 | 0 | 0.18 | 0.4 | 0 | 0 | 0.06 | 0 | 0.18 | 0.3 | 0 | 0 | 0.06 | 0 | 0.18 | 0.4 | Ext. | Ext. | Ext. | - | - | Ext. | - | - | Ext. |
| | S | 0.24 | 2.3 | 0.9 | 17.7 | 2.71 | 29.5 | 0.25 | 2.4 | 1.01 | 19.1 | 2.76 | 30.8 | 0.23 | 2.3 | 0.9 | 17.7 | 2.7 | 29.5 | Ext. | Ext. | Ext. | Ext. | Ext. | Ext. | Ext. | Ext. | Ext. |
| Darwin | N | 0.35 | 2.2 | 0.94 | 2.8 | 1.43 | 6.3 | 0.37 | 2.3 | 0.94 | 3 | 1.41 | 6.5 | 0.34 | 2.2 | 0.94 | 2.8 | 1.44 | 6.4 | Ext. | Ext. | Ext. | Ext. | Ext. | Ext. | Ext. | Ext. | Ext. |
| | S | 0.01 | 0 | 0.12 | 0.6 | 0.37 | 1.4 | 0.01 | 0 | 0.12 | 0.7 | 0.35 | 1.4 | 0.01 | 0 | 0.13 | 0.7 | 0.37 | 1.5 | Ext. | Ext. | Ext. | - | Ext. | Ext. | - | Ext. | Ext. |
| Brisbane | N | 0 | 0 | 0.01 | 0 | 0.07 | 0 | 0 | 0 | 0.01 | 0 | 0.07 | 0 | 0.01 | 0 | 0.01 | 0 | 0.07 | 0 | Ext. | Ext. | Ext. | - | - | - | - | - | - |
| | S | 0.29 | 6.6 | 0.3 | 6.5 | 0.3 | 6.5 | 0.03 | 0 | 0.09 | 0.1 | 0.19 | 2.7 | 0.11 | 1.3 | 0.11 | 1.2 | 0.17 | 2.3 | Int. | Int. | Int. | - | Ext. | Ext. | Int. | Int. | Ext. |
| Melbourne | N | 0.12 | 3.8 | 0.12 | 3.5 | 0.11 | 3.4 | 0 | 0 | 0 | 0 | 0 | 0 | 0 | 0 | 0 | 0 | 0 | 0 | Int. | Int. | Int. | - | - | - | - | - | - |
| | S | 3.45 | 62.2 | 3.43 | 61.5 | 3.42 | 61.4 | 0.13 | 3.5 | 0.11 | 3 | 0.1 | 2.9 | 2.66 | 55.2 | 2.61 | 54.3 | 2.6 | 54.1 | Int. | Int. | Int. | Int. | Int. | Int. | Int. | Int. | Int. |

## Appendix B. Summary of Unprotected CLT Results

**Table A4.** Unprotected CLT results for external insulation. O (orientation). Performance criteria: A (mould growth index), B (corrosion rate). Classification of WRB: Class 2, 3, and 4 (vapour permeance). Green (pass), red (fail), or yellow (further investigation). Critical surface that fails the performance criteria indicated in right table: Ext. (external surface), Int. (internal surface), Both (both surfaces).

**Unprotected Case 1**

| | | Mineral Wool No WRB | | EPS No WRB | | Wood Fibre No WRB | | Critical Surface Mineral Wool No WRB | EPS No WRB | Wood Fibre No WRB |
|---|---|---|---|---|---|---|---|---|---|---|
| Climate | O | A | B | A | B | A | B | | | |
| Cairns | N | 1.73 | 6.1 | 2.34 | 14 | 1.28 | 5.5 | Ext. | Ext. | Ext. |
| | S | 3.76 | 48.1 | 2.94 | 26 | 3.63 | 48.5 | Ext. | Ext. | Ext. |
| Darwin | N | 3.2 | 31.7 | 2.61 | 13.7 | 3.67 | 27.4 | Ext. | Ext. | Ext. |
| | S | 2.43 | 26.2 | 2.76 | 17.2 | 1.71 | 16.6 | Ext. | Ext. | Ext. |
| Brisbane | N | 0.86 | 1.7 | 2.18 | 14.4 | 0.41 | 2 | Ext. | Ext. | Ext. |
| | S | 1.45 | 6.2 | 2.94 | 26.7 | 1.04 | 5.3 | Ext. | Ext. | Ext. |
| Melbourne | N | 0.2 | 0.4 | 2.72 | 21.8 | 0.14 | 0 | Ext. | Ext. | - |
| | S | 0.53 | 1.4 | 3.06 | 30.2 | 0.15 | 0.5 | Ext. | Ext. | Both |

**Unprotected Case 2 — Performance Criteria**

| | | Mineral Wool Class 2 | | Class 3 | | Class 4 | | EPS Class 2 | | Class 3 | | Class 4 | | Wood Fibre Class 2 | | Class 3 | | Class 4 | |
|---|---|---|---|---|---|---|---|---|---|---|---|---|---|---|---|---|---|---|---|
| Climate | O | A | B | A | B | A | B | A | B | A | B | A | B | A | B | A | B | A | B |
| Cairns | N | 2.57 | 15.5 | 1.9 | 7.3 | 1.78 | 6.2 | 2.32 | 14 | 2.38 | 14.3 | 2.32 | 14 | 1.8 | 6.2 | 1.97 | 7.9 | 1.8 | 6.2 |
| | S | 2.99 | 31.7 | 2.98 | 43.8 | 3.39 | 47.7 | 3.05 | 26 | 2.96 | 25.3 | 2.92 | 25.6 | 3.02 | 35.6 | 2.97 | 45.4 | 3.29 | 48.3 |
| Darwin | N | 2.63 | 13.4 | 2.67 | 19.6 | 3.03 | 30.6 | 2.62 | 14.3 | 2.59 | 13.7 | 2.59 | 13.6 | 2.79 | 13.8 | 3.15 | 17 | 3.8 | 24 |
| | S | 2.67 | 14.9 | 2.38 | 17.7 | 2.36 | 22.8 | 2.9 | 18.6 | 2.79 | 17.5 | 2.75 | 17 | 2.72 | 15.2 | 2.38 | 13.4 | 2.29 | 16.7 |
| Brisbane | N | 2.02 | 13.4 | 1.32 | 3.1 | 1 | 2 | 2.55 | 21.6 | 2.29 | 15.3 | 2.18 | 14.4 | 2.08 | 13.7 | 1.47 | 3.6 | 1.12 | 2.9 |
| | S | 2.85 | 25.6 | 1.98 | 15.1 | 1.57 | 6.9 | 3.14 | 29.7 | 2.99 | 27.5 | 2.93 | 26.6 | 2.9 | 26.2 | 2.08 | 15.8 | 1.65 | 7.8 |
| Melbourne | N | 2.05 | 15.8 | 1.52 | 9.8 | 0.32 | 0.7 | 2.81 | 25.6 | 2.57 | 19.9 | 2.44 | 18.8 | 2.25 | 17 | 1.13 | 3.1 | 0.68 | 1.9 |
| | S | 2.69 | 20.1 | 1.55 | 9.9 | 0.65 | 1.6 | 3.28 | 38.6 | 3.01 | 29.3 | 2.92 | 27.5 | 2.84 | 21.9 | 1.46 | 8.9 | 0.99 | 2.7 |

**Unprotected Case 2 — Critical Surface**

| Climate | O | MW Cla. 2 | Cla. 3 | Cla. 4 | EPS Cla. 2 | Cla. 3 | Cla. 4 | WF Cla. 2 | Cla. 3 | Cla. 4 |
|---|---|---|---|---|---|---|---|---|---|---|
| Cairns | N | Ext. | Ext. | Ext. | Ext. | Ext. | Ext. | Ext. | Ext. | Ext. |
| | S | Ext. | Ext. | Ext. | Ext. | Ext. | Ext. | Ext. | Ext. | Ext. |
| Darwin | N | Ext. | Ext. | Ext. | Ext. | Ext. | Ext. | Ext. | Ext. | Ext. |
| | S | Ext. | Ext. | Ext. | Ext. | Ext. | Ext. | Ext. | Ext. | Ext. |
| Brisbane | N | Ext. | Ext. | Ext. | Ext. | Ext. | Ext. | Ext. | Ext. | Ext. |
| | S | Ext. | Ext. | Ext. | Ext. | Ext. | Ext. | Ext. | Ext. | Ext. |
| Melbourne | N | Ext. | Int. | Ext. | Ext. | Ext. | Ext. | Ext. | Ext. | Ext. |
| | S | Ext. | Int. | Ext. | Ext. | Ext. | Ext. | Ext. | Ext. | Ext. |

**Unprotected Case 3 — Performance Criteria**

| | | Mineral Wool Class 2 | | Class 3 | | Class 4 | | EPS Class 2 | | Class 3 | | Class 4 | | Wood Fibre Class 2 | | Class 3 | | Class 4 | |
|---|---|---|---|---|---|---|---|---|---|---|---|---|---|---|---|---|---|---|---|
| Climate | O | A | B | A | B | A | B | A | B | A | B | A | B | A | B | A | B | A | B |
| Cairns | N | 2.65 | 14 | 1.95 | 7.5 | 1.79 | 6.4 | 2.55 | 15.4 | 2.4 | 14.2 | 2.36 | 14 | 1.55 | 12 | 1.28 | 6.1 | 1.27 | 5.5 |
| | S | 3.02 | 29.1 | 2.96 | 42.8 | 3.11 | 47.3 | 3.08 | 28 | 3 | 29.1 | 2.99 | 29.3 | 2.64 | 20.1 | 2.55 | 36.9 | 2.69 | 45.2 |
| Darwin | N | 2.75 | 13.5 | 2.71 | 20.6 | 2.93 | 31.2 | 2.67 | 14.7 | 2.66 | 14 | 2.67 | 13.9 | 1.71 | 12 | 1.81 | 13 | 2.17 | 22.5 |
| | S | 2.8 | 15.4 | 2.53 | 19.3 | 2.43 | 25.4 | 2.93 | 19 | 2.83 | 18 | 2.8 | 17.6 | 2.08 | 13.3 | 1.56 | 11.4 | 1.56 | 15.8 |
| Brisbane | N | 2.47 | 13.6 | 1.42 | 3.7 | 1.06 | 2.1 | 2.58 | 21.9 | 2.33 | 15.5 | 2.25 | 15.1 | 1.34 | 11 | 0.76 | 3.2 | 0.51 | 2.6 |
| | S | 2.96 | 26 | 2.17 | 15.3 | 1.67 | 7.5 | 3.16 | 30.3 | 3.01 | 27.7 | 2.96 | 27.1 | 2.37 | 19.9 | 1.33 | 15.3 | 1.12 | 7.1 |
| Melbourne | N | 2.4 | 16.9 | 0.8 | 2 | 0.35 | 0.9 | 2.93 | 27.5 | 2.78 | 22.6 | 2.74 | 22 | 0.79 | 8 | 0.14 | 0 | 0.14 | 0 |
| | S | 3.28 | 20.6 | 1.07 | 4.4 | 0.55 | 1.5 | 3.28 | 39.6 | 3.12 | 31.7 | 3.08 | 31 | 1.35 | 10.6 | 0.15 | 0.3 | 0.15 | 0.1 |

**Unprotected Case 3 — Critical Surface**

| Climate | O | MW Cla. 2 | Cla. 3 | Cla. 4 | EPS Cla. 2 | Cla. 3 | Cla. 4 | WF Cla. 2 | Cla. 3 | Cla. 4 |
|---|---|---|---|---|---|---|---|---|---|---|
| Cairns | N | Ext. | Ext. | Ext. | Ext. | Ext. | Ext. | Ext. | Ext. | Ext. |
| | S | Ext. | Ext. | Ext. | Ext. | Ext. | Ext. | Ext. | Ext. | Ext. |
| Darwin | N | Ext. | Ext. | Ext. | Ext. | Ext. | Ext. | Ext. | Ext. | Ext. |
| | S | Ext. | Ext. | Ext. | Ext. | Ext. | Ext. | Ext. | Ext. | Ext. |
| Brisbane | N | Ext. | Ext. | Ext. | Ext. | Ext. | Ext. | Ext. | Ext. | Ext. |
| | S | Ext. | Ext. | Ext. | Ext. | Ext. | Ext. | Ext. | Ext. | Ext. |
| Melbourne | N | Ext. | Ext. | Ext. | Ext. | Ext. | Ext. | Ext. | - | - |
| | S | Ext. | Ext. | Ext. | Ext. | Ext. | Ext. | Ext. | Both | Int. |

**Table A5.** Unprotected CLT results for split insulation. O (orientation). Performance criteria: A (mould growth index), B (corrosion rate). Classification of WRB: Class 2, 3, and 4 (vapour permeance). Green (pass), red (fail), or yellow (further investigation). Critical surface that fails the performance criteria indicated in right table: Ext. (external surface), Int. (internal surface), Both (both surfaces).

**Unprotected Case 1**

| Climate | O | Mineral Wool No WRB A | B | EPS No WRB A | B | Wood Fibre No WRB A | B | Mineral Wool No WRB | EPS No WRB | Wood Fibre No WRB |
|---|---|---|---|---|---|---|---|---|---|---|
| Cairns | N | 1.36 | 3.6 | 2.09 | 11 | 1.2 | 3.9 | Ext. | Int. | Ext. |
| | S | 3.86 | 39.3 | 2.7 | 29.2 | 3.97 | 43.3 | Ext. | Ext. | Ext. |
| Darwin | N | 3 | 16.5 | 2.29 | 10.6 | 4.12 | 17.8 | Ext. | Ext. | Ext. |
| | S | 2 | 10 | 2.34 | 15.3 | 1.69 | 9.4 | Ext. | Ext. | Ext. |
| Brisbane | N | 0.55 | 1 | 2.3 | 18.8 | 0.32 | 1.3 | Ext. | Int. | Ext. |
| | S | 1.22 | 6.6 | 2.51 | 25.6 | 1.04 | 6.3 | Ext. | Both | Ext. |
| Melbourne | N | 0.21 | 0.4 | 2.34 | 27.3 | 0.05 | 0 | Int. | Int. | - |
| | S | 0.98 | 25.2 | 2.81 | 37.5 | 0.45 | 6.6 | Int. | Ext. | Int. |

**Unprotected Case 2**

| Climate | O | MW Cl2 A | B | MW Cl3 A | B | MW Cl4 A | B | EPS Cl2 A | B | EPS Cl3 A | B | EPS Cl4 A | B | WF Cl2 A | B | WF Cl3 A | B | WF Cl4 A | B | MW Cla.2 | MW Cla.3 | MW Cla.4 | EPS Cla.2 | EPS Cla.3 | EPS Cla.4 | WF Cla.2 | WF Cla.3 | WF Cla.4 |
|---|---|---|---|---|---|---|---|---|---|---|---|---|---|---|---|---|---|---|---|---|---|---|---|---|---|---|---|---|
| Cairns | N | 1.44 | 3.6 | 1.63 | 3.5 | 1.44 | 3.6 | 2.04 | 9.8 | 2.04 | 12 | 2.04 | 9.8 | 1.52 | 3.9 | 1.72 | 4 | 1.52 | 3.9 | Ext. | Ext. | Ext. | Int. | Int. | Int. | Ext. | Ext. | Ext. |
| | S | 2.87 | 23.3 | 2.87 | 37.4 | 3.29 | 40.2 | 3.01 | 34.2 | 2.74 | 30.7 | 2.71 | 28.9 | 2.94 | 29.6 | 2.94 | 41 | 3.52 | 43.8 | Ext. | Ext. | Ext. | Ext. | Ext. | Ext. | Ext. | Ext. | Ext. |
| Darwin | N | 2.33 | 12.6 | 2.46 | 11 | 2.82 | 15.4 | 2.41 | 13.8 | 2.31 | 13.2 | 2.27 | 10.8 | 2.59 | 13 | 2.95 | 12.6 | 3.76 | 16.9 | Ext. | Ext. | Ext. | Ext. | Ext. | Ext. | Ext. | Ext. | Ext. |
| | S | 2.52 | 13.7 | 2.08 | 9.6 | 2.02 | 8.8 | 2.74 | 22.3 | 2.48 | 17.5 | 2.35 | 15.4 | 2.59 | 14.2 | 2.16 | 10.9 | 2.06 | 8.5 | Ext. | Ext. | Ext. | Ext. | Ext. | Ext. | Ext. | Ext. | Ext. |
| Brisbane | N | 1.74 | 10.6 | 1.11 | 2.3 | 0.69 | 1.5 | 2.26 | 22.5 | 2.26 | 19.1 | 2.26 | 18.8 | 1.8 | 11.6 | 1.2 | 2.8 | 0.83 | 1.8 | Ext. | Ext. | Ext. | Both | Int. | Int. | Ext. | Ext. | Ext. |
| | S | 2.83 | 26.2 | 1.88 | 14.1 | 1.36 | 7.1 | 3.03 | 38.4 | 2.72 | 28.5 | 2.55 | 25.7 | 2.85 | 26.5 | 1.94 | 15.2 | 1.45 | 7.5 | Ext. | Ext. | Ext. | Ext. | Ext. | Ext. | Ext. | Ext. | Ext. |
| Melbourne | N | 1.94 | 15.7 | 1.7 | 10.4 | 0.23 | 0.4 | 2.75 | 31.7 | 2.99 | 30.2 | 2.31 | 26.9 | 2.09 | 16.7 | 0.94 | 2.3 | 0.47 | 1.2 | Ext. | Int. | Ext. | Ext. | Int. | Int. | Ext. | Ext. | Ext. |
| | S | 2.75 | 26.9 | 2.21 | 19 | 0.95 | 24.8 | 3.21 | 59.2 | 2.97 | 41.2 | 2.81 | 37.5 | 2.85 | 27.5 | 1.37 | 8.5 | 0.86 | 7.6 | Ext. | Int. | Int. | Ext. | Int. | Ext. | Ext. | Ext. | Both |

**Unprotected Case 3**

| Climate | O | MW Cl2 A | B | MW Cl3 A | B | MW Cl4 A | B | EPS Cl2 A | B | EPS Cl3 A | B | EPS Cl4 A | B | WF Cl2 A | B | WF Cl3 A | B | WF Cl4 A | B | MW Cla.2 | MW Cla.3 | MW Cla.4 | EPS Cla.2 | EPS Cla.3 | EPS Cla.4 | WF Cla.2 | WF Cla.3 | WF Cla.4 |
|---|---|---|---|---|---|---|---|---|---|---|---|---|---|---|---|---|---|---|---|---|---|---|---|---|---|---|---|---|
| Cairns | N | 2.08 | 12.5 | 1.67 | 3.3 | 1.46 | 3.5 | 2.22 | 14.9 | 2.08 | 13.2 | 2.08 | 12.1 | 1.58 | 10.1 | 1.32 | 3.4 | 1.23 | 3.6 | Ext. | Ext. | Ext. | Both | Int. | Int. | Ext. | Ext. | Ext. |
| | S | 2.92 | 23.6 | 2.83 | 35.9 | 2.99 | 39.6 | 2.98 | 35.3 | 2.84 | 32.8 | 2.83 | 34.7 | 2.74 | 20 | 2.59 | 33.8 | 2.7 | 41.4 | Ext. | Ext. | Ext. | Ext. | Ext. | Ext. | Ext. | Ext. | Ext. |
| Darwin | N | 2.4 | 12.6 | 2.44 | 10.7 | 2.69 | 16.5 | 2.5 | 13.9 | 2.46 | 13.2 | 2.44 | 11.6 | 1.78 | 10.1 | 1.9 | 8.3 | 2.17 | 15.7 | Ext. | Ext. | Ext. | Ext. | Ext. | Ext. | Ext. | Ext. | Ext. |
| | S | 2.7 | 14 | 2.08 | 10.3 | 1.97 | 10 | 2.77 | 22.6 | 2.54 | 17.9 | 2.46 | 17.1 | 2.12 | 13.4 | 1.56 | 9.8 | 1.58 | 8.3 | Ext. | Ext. | Ext. | Ext. | Ext. | Ext. | Ext. | Ext. | Ext. |
| Brisbane | N | 1.87 | 11.1 | 1.16 | 2.6 | 0.77 | 1.6 | 2.3 | 19.8 | 2.3 | 19.2 | 2.3 | 19 | 1.33 | 9.1 | 0.79 | 2.6 | 0.48 | 1.8 | Ext. | Ext. | Ext. | Int. | Int. | Int. | Ext. | Ext. | Ext. |
| | S | 2.93 | 26.9 | 2.09 | 15 | 1.48 | 7.9 | 3.05 | 42 | 2.74 | 28.8 | 2.64 | 27.1 | 2.69 | 24.5 | 1.57 | 15.1 | 1.22 | 8 | Ext. | Ext. | Ext. | Ext. | Ext. | Ext. | Ext. | Ext. | Ext. |
| Melbourne | N | 2.29 | 16.6 | 0.68 | 1.7 | 0.26 | 0.5 | 2.76 | 31.7 | 2.43 | 28.1 | 2.34 | 27.5 | 1.06 | 11 | 0.05 | 0 | 0.05 | 0 | Ext. | Ext. | Ext. | Ext. | Both | Int. | Ext. | - | - |
| | S | 3.37 | 29.5 | 1.07 | 24 | 0.94 | 23.6 | 3.21 | 59.2 | 2.94 | 40 | 2.85 | 37.8 | 2.1 | 20.1 | 0.43 | 6.5 | 0.44 | 6.4 | Ext. | Both | Int. | Ext. | Ext. | Ext. | Ext. | Int. | Int. |

**Table A6.** Unprotected CLT results for internal insulation. O (orientation). Performance criteria: A (mould growth index), B (corrosion rate). Classification of WRB: Class 2, 3, and 4 (vapour permeance). Green (pass), red (fail), or yellow (further investigation). Critical surface that fails the performance criteria indicated in right table: Ext. (external surface), Int. (internal surface), Both (both surfaces).

| Unprotected Case 1 | | Mineral Wool No WRB | | EPS No WRB | | Wood Fibre No WRB | | Mineral Wool No WRB | EPS No WRB | Wood Fibre No WRB |
|---|---|---|---|---|---|---|---|---|---|---|
| Climate | O | A | B | A | B | A | B | | | |
| Cairns | N | 1 | 2.6 | 2.74 | 15.8 | 0.99 | 2.6 | Ext. | Int. | Ext. |
| | S | 4.13 | 35.9 | 4.16 | 38.1 | 4.11 | 35.8 | Ext. | Ext. | Ext. |
| Darwin | N | 2.44 | 8.8 | 2.74 | 15.9 | 2.43 | 8.9 | Ext. | Int. | Ext. |
| | S | 1.3 | 2.7 | 2.62 | 16.4 | 1.29 | 2.7 | Ext. | Int. | Ext. |
| Brisbane | N | 0.29 | 0.4 | 2.81 | 19.2 | 0.28 | 0.4 | Ext. | Int. | Ext. |
| | S | 1.14 | 9.2 | 2.76 | 28.6 | 1.12 | 7.7 | Both | Int. | Ext. |
| Melbourne | N | 0.71 | 9.4 | 2.98 | 29.1 | 0.53 | 3.8 | Int. | Int. | Int. |
| | S | 3.62 | 66.9 | 2.97 | 48 | 2.91 | 60.6 | Int. | Int. | Int. |

| Unprotected Case 2 | | Mineral Wool Class 2 | | Class 3 | | Class 4 | | EPS Class 2 | | Class 3 | | Class 4 | | Wood Fibre Class 2 | | Class 3 | | Class 4 | | Mineral Wool Cla. 2 | Cla. 3 | Cla. 4 | EPS Cla. 2 | Cla. 3 | Cla. 4 | Wood Fibre Cla. 2 | Cla. 3 | Cla. 4 |
|---|---|---|---|---|---|---|---|---|---|---|---|---|---|---|---|---|---|---|---|---|---|---|---|---|---|---|---|---|
| Climate | O | A | B | A | B | A | B | A | B | A | B | A | B | A | B | A | B | A | B | | | | | | | | | |
| Cairns | N | 1.74 | 5.6 | 1.32 | 2.5 | 1.07 | 2 | 2.71 | 18.8 | 2.71 | 16.5 | 2.71 | 15.4 | 1.73 | 5.9 | 1.32 | 2.5 | 1.07 | 2 | Ext. | Ext. | Ext. | Int. | Int. | Int. | Ext. | Ext. | Ext. |
| | S | 2.71 | 17.7 | 2.36 | 25.9 | 2.86 | 33.1 | 2.73 | 34.3 | 2.62 | 31.1 | 3.01 | 35.9 | 2.71 | 18.2 | 2.39 | 25.9 | 2.87 | 33.1 | Ext. | Ext. | Ext. | Ext. | Both | Ext. | Ext. | Ext. | Ext. |
| Darwin | N | 2.01 | 3.6 | 2 | 3.8 | 2.17 | 7.1 | 2.71 | 17.3 | 2.71 | 15.7 | 2.7 | 15.2 | 2.01 | 5 | 2 | 3.9 | 2.17 | 7.2 | Ext. | Ext. | Ext. | Int. | Int. | Int. | Ext. | Ext. | Ext. |
| | S | 2.17 | 12.1 | 1.54 | 3.4 | 1.3 | 2.7 | 2.59 | 20.1 | 2.59 | 16.9 | 2.58 | 16.3 | 2.16 | 12.3 | 1.54 | 3.4 | 1.3 | 2.7 | Ext. | Ext. | Ext. | Int. | Int. | Int. | Ext. | Ext. | Ext. |
| Brisbane | N | 1.47 | 6.4 | 0.88 | 1.8 | 0.42 | 0.8 | 2.78 | 25.1 | 2.78 | 19.6 | 2.78 | 19.1 | 1.47 | 6.5 | 0.88 | 1.8 | 0.42 | 0.8 | Ext. | Ext. | Ext. | Int. | Int. | Int. | Ext. | Ext. | Ext. |
| | S | 2.72 | 26.6 | 1.76 | 13.3 | 1.16 | 9.1 | 2.74 | 40.5 | 2.73 | 29.7 | 2.73 | 28.5 | 2.73 | 26.8 | 1.77 | 13.5 | 1.16 | 8.1 | Ext. | Ext. | Both | Ext. | Int. | Int. | Ext. | Ext. | Ext. |
| Melbourne | N | 0.1 | 2.1 | 0.11 | 2.3 | 0.11 | 2.3 | 2.95 | 39.8 | 2.95 | 29.6 | 2.95 | 28.8 | 1.07 | 9 | 0.99 | 7.7 | 1 | 7.2 | Int. | Int. | Int. | Int. | Int. | Int. | Both | Int. | Int. |
| | S | 3.56 | 57.2 | 3.6 | 56.3 | 3.61 | 55.9 | 2.94 | 81.2 | 2.94 | 48.5 | 2.94 | 47.2 | 3.07 | 66.1 | 3.07 | 63.1 | 3.07 | 62.3 | Int. | Int. | Int. | Int. | Int. | Int. | Int. | Int. | Int. |

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
