# Peer review of "Mass Timber Envelopes in Passivhaus Buildings: Designing for Moisture Safety in Hot and Humid Australian Climates"

_buildings, doi:10.3390/buildings11100478_

Round 1
Reviewer 1 Report
Mass Timber Envelopes in Passivhaus Buildings: Designing for Moisture Safety in Hot and Humid Australian Climates
The topic of the article is very interesting. CLT is a modern material that is successfully used in construction all over the world. Research on determining the correct conditions for the use of this material, in particular wall structures based on CLT, are important both from a scientific and utilitarian point of view. The presented research methodology does not raise any objections, and the results are correctly presented and discussed. However, according to the reviewer, the article requires a few technical adjustments:
- Line 402, 403, 404, 405 – In Fig. 3, a description of the vertical axes should be entered
- Line 498, 499 – The statement "Table 5" should be moved to the previous line
- Line 638, 639 – Table 8 is divided into 3 separate tables. In the opinion of the reviewer, since it is one table, it should be merged or adequately described, e.g. table 8a, table 8b, table 8c.
- Line 699, 700 – Table 10, like Table 8, is divided into 3 separate tables. In the opinion of the reviewer, since it is one table, it should be merged or adequately described, e.g. table 10a, table 10b, table 10c.
- Line 730 – Table 12 is divided into 2 separate tables. In the opinion of the reviewer, since it is one table, it should be merged or adequately described, e.g. table 12a, table 12b.
- Line 929-963 - The tables in Appendix A and B, like tables 8, 10 and 12, should be merged or properly described.
Taking into account the above remarks, the article requires a minor revision.
Author Response
|
Reviewers Comment |
Authors Response |
|
Line 402, 403, 404, 405 – In Fig. 3, a description of the vertical axes should be entered |
Resolved |
|
Line 498, 499 – The statement "Table 5" should be moved to the previous line |
Resolved |
|
Line 638, 639 – Table 8 is divided into 3 separate tables. In the opinion of the reviewer, since it is one table, it should be merged or adequately described, e.g. table 8a, table 8b, table 8c. |
Resolved, tables merged. |
|
Line 699, 700 – Table 10, like Table 8, is divided into 3 separate tables. In the opinion of the reviewer, since it is one table, it should be merged or adequately described, e.g. table 10a, table 10b, table 10c. |
Resolved, tables merged. |
|
Line 730 – Table 12 is divided into 2 separate tables. In the opinion of the reviewer, since it is one table, it should be merged or adequately described, e.g. table 12a, table 12b. |
Resolved, tables merged. |
|
Line 929-963 - The tables in Appendix A and B, like tables 8, 10 and 12, should be merged or properly described. |
Resolved, tables merged. |

Reviewer 2 Report
The manuscript presents simulation results of cross-laminated timber assemblies submitted to hot and humid climate conditions. The study performed by the authors is comprehensive and the discussions on results are relevant.
The manuscript is well and clearly written and can be published after minor corrections:
- please indicate what means the porosity of materials having the unit m2/m3
- the sentence: "The performance criteria result for protected assemblies are presented in Table 10." (lines 688 and 689) is repeated with "Results for the protected assemblies with split insulation are shown in Table 10." (line 701)
- Line 707: instead of Table 9 it should be Table 11
Author Response
|
Reviewers Comment |
Authors Response |
|
please indicate what means the porosity of materials having the unit m2/m3 |
Porosity determines the maximum water content in the material. You had found an error, it should have been in units m³/m³, now updated. Did you want me to provide a definition for porosity in the journal? Should I include definitions for the other hygrothermal properties as well? Please confirm. |
|
the sentence: "The performance criteria result for protected assemblies are presented in Table 10." (lines 688 and 689) is repeated with "Results for the protected assemblies with split insulation are shown in Table 10." (line 701) |
Repeat deleted. |
|
Line 707: instead of Table 9 it should be Table 11 |
Table 9 reference updated to table 11. |

Reviewer 3 Report
The topic of the manuscript is really interesting and provides new knowledge. The whole manuscript is found analytical and is quite well prepared, though there are several issues that should be addressed before publication. Here are my comments and recommendations towards the improvement of this manuscript.
You should check again if the part of contents is appropriate to be added in the text. Did you present your manuscript following the instructions for the authors file? Please, provide references for the part of lines 167-181. In line 199, please provide some significant information on AIRAH DA20 technical manual. In the end of the text, funding information, even though very critical to the readers as well as the reviewers, is not clear. In some cases, you present one and in some other cases 2 decimals in the values of the tables. The colours Green (pass), Red (fail), and Yellow (further investigation) in table caption do not correspond to the same colours depicted in the table cells. A brief comment in my opinion is necessary to be added concerning the species used for CLT production applied on such climates. In conclusions please highlight the practical meaning and significance of the findings.
Author Response
|
Reviewers Comment |
Authors Response |
|
You should check again if the part of contents is appropriate to be added in the text. Did you present your manuscript following the instructions for the authors file? |
I had followed the instructions for the authors, it states ‘Buildings now accepts free format submission’. Therefore, I included the contents, though I was careful to include all required sections. |
|
Please, provide references for the part of lines 167-181. |
Resolved, reference copied from line 167 to line 181. This is because the whole paragraph refers to information from reference [34]. |
|
In line 199, please provide some significant information on AIRAH DA20 technical manual. |
Title of manual added. A number of other sentences in that paragraph had also been referencing the manual [35]. |
|
In the end of the text, funding information, even though very critical to the readers as well as the reviewers, is not clear. |
I’m not sure how the funding description could be clearer, please clarify? |
|
In some cases, you present one and in some other cases 2 decimals in the values of the tables. |
This is because the values with 1 decimal place refers to a zero in the second decimal place eg. 0.4 = 0.40. Please confirm if my method is fine for the given page space, or if I should remove all second decimal places for all values? As I do not have enough space on the page to include 2 decimal places for all values. |
|
The colours Green (pass), Red (fail), and Yellow (further investigation) in table caption do not correspond to the same colours depicted in the table cells. |
The colours in the table caption font were identical to the colours in the table’s font. I have now removed off the font bolding, as it may have altered the perception of the colour. |
|
A brief comment in my opinion is necessary to be added concerning the species used for CLT production applied on such climates. |
Please see line 499 to line 504 and confirm if further information is required concerning CLT species in the simulated climates. |
|
In conclusions please highlight the practical meaning and significance of the findings. |
Re-reading the ‘conclusion’ section (and my original ‘aims and scope’ section. I believe that the practical considerations and significant of findings are already implemented. Please clarify what additional significance of the findings should also be considered? |
Round 2
Reviewer 3 Report
The authors followed and implemented only some of the recommendations of the reviewer. Nevertheless, I believe that the most significant issues have been addressed. Therefore, I leave the final decision to the Editors.
Author Response
Thank you very much for your reviews.
- Some terminology in the funding section has been altered, as per the editors clarification.
- No alteration has been made to the decimal places. The editor has accepted the current format as being acceptable.
- Several sentences have been added to the conclusion to highlight the practical implementation of the findings.